# SIRT7 suppresses energy expenditure and thermogenesis by regulating brown adipose tissue functions in mice

Tatsuya Yoshizawa [1] ✉, Yoshifumi Sato [1], Shihab U. Sobuz[1], Tomoya Mizumoto[1], Tomonori Tsuyama[2], Md. Fazlul Karim[1], Keishi Miyata[3], Masayoshi Tasaki[4,5], Masaya Yamazaki [1], Yuichi Kariba[1,6], Norie Araki[7], Eiichi Araki[2,6], Shingo Kajimura [8], Yuichi Oike [2,3], Thomas Braun [9], Eva Bober[9], Johan Auwerx [10] & Kazuya Yamagata[1,2] ✉

Brown adipose tissue plays a central role in the regulation of the energy balance by expending energy to produce heat. NAD⁺-dependent deacylase sirtuins have widely been recognized as positive regulators of brown adipose tissue thermogenesis. However, here we reveal that SIRT7, one of seven mammalian sirtuins, suppresses energy expenditure and thermogenesis by regulating brown adipose tissue functions. Whole-body and brown adipose tissue-specific *Sirt7* knockout mice have higher body temperature and energy expenditure. SIRT7 deficiency increases the protein level of UCP1, a key regulator of brown adipose tissue thermogenesis. Mechanistically, we found that SIRT7 deacetylates insulin-like growth factor 2 mRNA-binding protein 2, an RNA-binding protein that inhibits the translation of *Ucp1* mRNA, thereby enhancing its inhibitory action on *Ucp1*. Furthermore, SIRT7 attenuates the expression of batokine genes, such as *fibroblast growth factor 21*. In conclusion, we propose that SIRT7 serves as an energy-saving factor by suppressing brown adipose tissue functions.

Mammals maintain energy homeostasis by balancing energy intake and expenditure. Thermogenesis is a major homeostatic mechanism for dissipating energy through the production of heat, and brown adipose tissue (BAT) plays a crucial role in whole-body energy homeostasis through non-shivering thermogenesis[1]. Two types of thermogenic adipocytes—classical brown adipocytes and beige adipocytes—are present in mammals. Classical brown adipocytes constitutively

reside in dedicated BAT depots such as the interscapular regions, and beige adipocytes can be induced within white adipose tissue (WAT) depots in response to a variety of environmental stimuli, including chronic cold acclimation and exercise[2,3]. Recent studies have demonstrated that the activity/prevalence of adult human BAT is inversely correlated with body mass index and glucose level[4–9]. Therefore, activation of BAT may be an attractive therapeutic target for the

[1]Department of Medical Biochemistry, Faculty of Life Sciences, Kumamoto University, Kumamoto 860-8556, Japan. [2]Center for Metabolic Regulation of Healthy Aging, Faculty of Life Sciences, Kumamoto University, Kumamoto 860-8556, Japan. [3]Department of Molecular Genetics, Faculty of Life Sciences, Kumamoto University, Kumamoto 860-8556, Japan. [4]Department of Neurology, Faculty of Life Sciences, Kumamoto University, Kumamoto 860-8556, Japan. [5]Department of Biomedical Laboratory Sciences, Faculty of Life Sciences, Kumamoto University, Kumamoto 860-8556, Japan. [6]Department of Metabolic Medicine, Faculty of Life Sciences, Kumamoto University, Kumamoto 860-8556, Japan. [7]Department of Tumor Genetics and Biology, Faculty of Life Sciences, Kumamoto University, Kumamoto 860-8556, Japan. [8]Beth Israel Deaconess Medical Center, Harvard Medical School, and Howard Hughes Medical Institute, Boston, MA 02215, USA. [9]Department of Cardiac Development and Remodeling, Max-Planck-Institute for Heart and Lung Research, Ludwigstr. 43, 61231 Bad Nauheim, Germany. [10]Laboratory of Integrative Systems Physiology, Interfaculty Institute of Bioengineering, École Polytechnique Fédérale de Lausanne, Lausanne, Switzerland. ✉e-mail: yoshizaw@kumamoto-u.ac.jp; k-yamaga@kumamoto-u.ac.jp

Nature Communications | (2022)13:7439 1

treatment of obesity and metabolic syndrome[10,11]. On the other hand, suppression of BAT thermogenesis is also quite important under energy-saving conditions, such as sleep or starvation[12–14].

The unique thermogenic capacity of BAT is attributable to the expression of uncoupling protein 1 (UCP1), which is a mitochondrial inner membrane protein capable of uncoupling the electrochemical gradient from ATP synthesis and dissipating energy as heat. Although some UCP1-independent thermogenic pathways have been reported, the adaptive thermogenic property of BAT relies predominantly on the actions of UCP1, as confirmed in genetically modified mice[15]. *Ucp1* knockout (KO) mice are cold-sensitive and susceptible to obesity at thermoneutrality[16,17], and overexpression of *Ucp1* in fat prevents genetic obesity[18]. Moreover, −3826A/G polymorphisms in the human *Ucp1* promoter influence its promoter activity and are reported to be associated with body fat accumulation in particular populations[19]. Accordingly, the molecular mechanism of *Ucp1* gene transcriptional control has been extensively investigated, leading to the discovery of many transcription factors, cofactors, and epigenetic factors[20]. In addition, studies are now gradually unraveling post-transcriptional regulatory processes, such as the splicing and translation of *Ucp1* mRNA[21–24].

Sirtuins (SIRT1–7 in mammals) are nicotinamide adenine dinucleotide (NAD⁺)-dependent lysine deacetylases/deacylases that regulate a wide variety of biological functions, including energy metabolism, stress resistance, tumorigenesis, and aging[25–27]. Several sirtuins have been reported to be positive regulators of BAT thermogenesis. For example, fat-specific *Sirt6* KO mice show reduced energy expenditure and lower body temperature due to impaired BAT thermogenic function[28]. *Sirt1* heterozygous KO mice exhibit increased adiposity and more severe insulin resistance with less thermogenesis under high-fat diet (HFD) challenge[29]. *Sirt5* KO mice on a Sv129 background show less browning (differentiation of beige adipocytes in WAT) capacity in WAT[30], and *Sirt5* KO in BAT leads to mitochondrial metabolic inflexibility and cold intolerance under fasting conditions[31]. *Sirt3* KO mice have impaired BAT lipid use and thermoregulation upon cold exposure[32].

Until recently, SIRT7 had been the least studied sirtuin, but it has lately been reported to have many important biological functions, with roles in ribosome biogenesis, the stress response, genome integrity, metabolism, cancer, and aging[33,34]. We previously found that *Sirt7* KO mice are resistant to HFD-induced obesity, glucose intolerance, and fatty liver, in striking contrast to other reported *sirtuin* KO mice[35]. We also found an increased body temperature in *Sirt7* KO mice along with increased expression of *Ucp1* in BAT under HFD conditions. Our findings suggested that SIRT7 may regulate whole-body energy metabolism by regulating lipid consumption in BAT. However, the precise functions and molecular mechanisms of SIRT7 in BAT thermogenesis and energy expenditure were unclear.

In the present study, we show the physiological roles of SIRT7 in BAT functions, with several lines of *Sirt7* deficient mice exhibiting impaired ability to decrease their body temperature and energy expenditure. We also demonstrate acetylation-dependent regulatory mechanisms attenuating the translation of *Ucp1* mRNA by insulin-like growth factor 2 mRNA-binding protein 2 (IGF2BP2/IMP2). Our findings establish SIRT7 as a critical energy-saving factor that suppresses BAT thermogenesis.

## Results

### SIRT7 deficiency enhances energy expenditure and thermogenesis under normal nutritional conditions

First, we examined whether SIRT7 deficiency under normal chow feeding conditions has any effect on energy expenditure and BAT thermogenesis. Under normal nutritional conditions, although the body weight of 15-week-old *Sirt7* KO mice was almost the same as that of wild-type (WT) control mice (Fig. 1a), the ratio of visceral epididymal

WAT (epiWAT) to body weight was lower (Fig. 1b). Indirect calorimetry analysis showed that oxygen consumption (VO₂), carbon dioxide production (VCO₂), and energy expenditure were significantly enhanced in *Sirt7* KO mice (Fig. 1c–f), whereas the respiratory exchange ratio (RER), locomotor activity, and water intake were unchanged in *Sirt7* KO mice (Fig. 1g–i). The slight increase in food intake versus body weight may have been caused by excessive energy consumption (Fig. 1j) because a previous report indicated that mice housed under ambient temperature (22 °C) increased food intake to meet the elevated energy demanded for adaptive thermogenesis compared with the thermoneutral condition (30 °C)[36].

Given that BAT plays a central role in controlling energy expenditure through thermogenesis, we next investigated BAT in *Sirt7* KO mice. The ratio of interscapular BAT (iBAT) to body weight was lower in *Sirt7* KO mice than in WT mice (Fig. 1b) and, consistently, *Sirt7* KO mice accumulated less lipid droplets in iBAT compared with WT mice on histological analysis (Fig. 1k). The body temperature of *Sirt7* KO mice was significantly higher than that of WT mice, especially in the light phase (Fig. 1l). Upon cold exposure at 4 °C, *Sirt7* KO mice displayed a modestly but significantly higher body temperature compared with WT mice (Supplementary Fig. 1a). In the case of female *Sirt7* KO mice, VO₂ and energy expenditure in the dark phase only and body temperature were similarly increased (Supplementary Fig. 1b–d). To confirm these phenotypes, we analyzed another independent line of *Sirt7* KO mice[37]. As shown in Supplementary Fig. 1e–g, at 2 months old, this *Sirt7* KO mouse line also exhibited higher VO₂, energy expenditure, and body temperature compared with WT mice. These results strongly indicate that SIRT7 suppresses whole-body energy expenditure and thermogenesis under normal nutritional conditions.

### Loss of SIRT7 impairs the ability of mice to decrease their energy expenditure and thermogenesis in the hypometabolic state

Aging is accompanied by reduced BAT thermogenesis and energy expenditure. Although the mechanisms of these metabolic changes in elderly people are poorly understood, a mechanism to actively suppress energy expenditure and BAT thermogenic function during aging has recently been identified[38]. Therefore, to investigate the role of SIRT7 in BAT functions during aging, we analyzed the thermogenic functions of 2-year-old WT and *Sirt7* KO mice. As shown in Fig. 2a–c and Supplementary Fig. 2a–d, VO₂, energy expenditure, and body temperature were significantly increased in *Sirt7* KO mice, whereas the RER, locomotor activity, and water intake were unchanged, while food intake versus body weight was slightly increased. VO₂, energy expenditure, and body temperature clearly declined in WT mice during aging but only mildly decreased in *Sirt7* KO mice, enlarging the difference between WT and *Sirt7* KO mice (Fig. 2a–c). These data imply that SIRT7 persistently serves as a suppressor of BAT thermogenesis, while overall thermogenesis gradually decreases during aging. As expected, in contrast to genes activating BAT thermogenesis—PR domain containing 16 (*Prdm16*); peroxisome proliferative activated receptor, gamma, coactivator 1 alpha (*Ppargc1a*); *Ucp1*; and calsyntenin 3ß (*Clstn3b*)—that exhibited attenuated expression in iBAT of aged mice, the expression of *Sirt7* mRNA was steady and SIRT7 protein was increased by aging (Supplementary Fig. 2e, f). These results suggest that SIRT7 contributes more substantially to the suppression of energy expenditure and thermogenesis in aged mice.

Mammals have acquired a mechanism to actively reduce thermogenesis during sleep or starvation, and some mammals enter an active hypometabolic state to survive, either daily torpor (minutes to hours) or hibernation (days to weeks)[39–41]. C57BL/6j laboratory mice can also enter daily torpor when they are fasted under cool ambient temperature[42]. Therefore, we investigated whether SIRT7 is linked to the hypometabolic state in daily torpor. First, we confirmed the torpor induction method with C57BL/6j mice. When C57BL/6j mice were placed in a metabolic chamber at 20 °C and fasted to induce torpor,

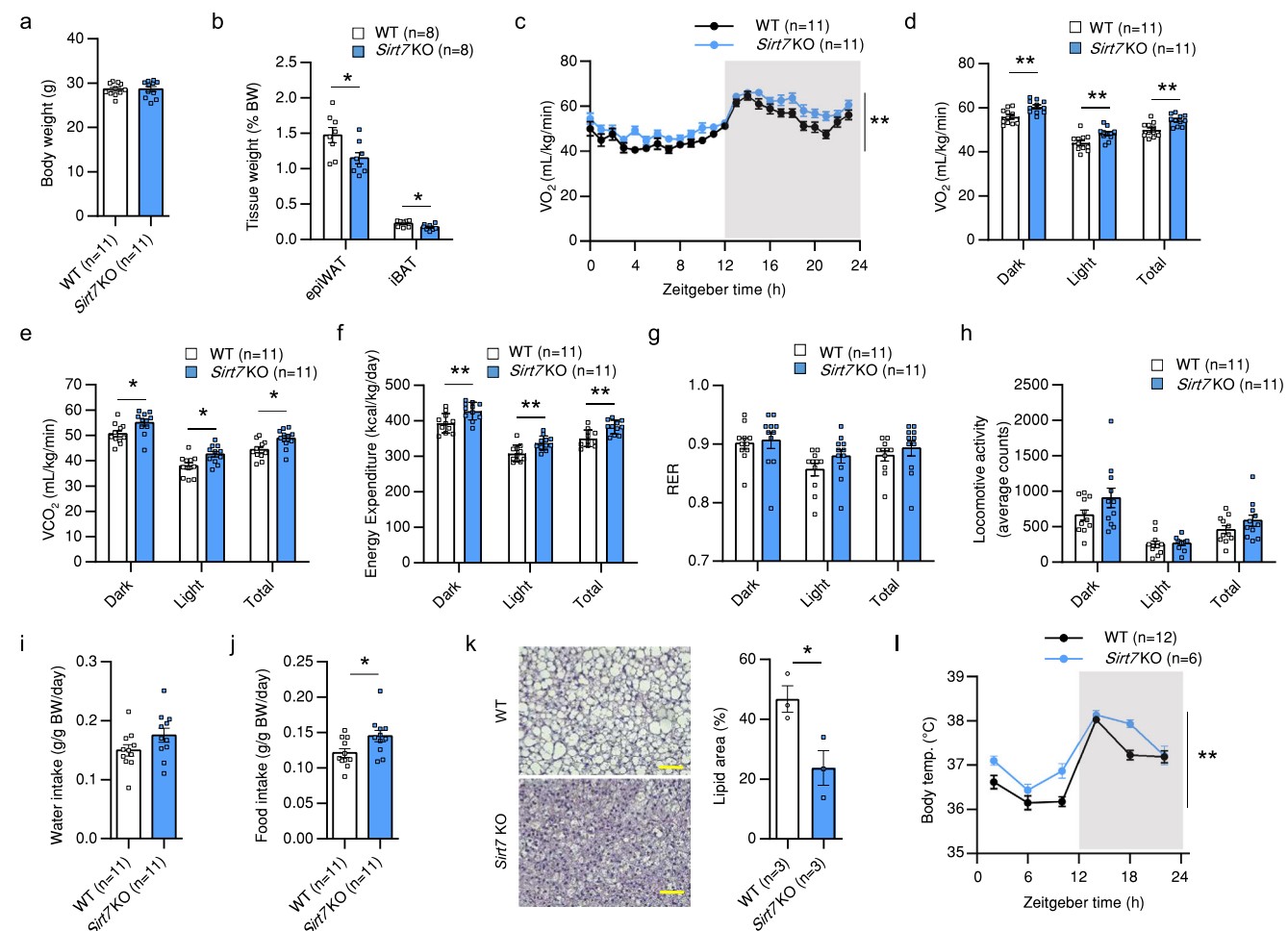

**Fig. 1 | *Sirt7* KO mice display elevated energy expenditure and body temperature under normal conditions. a, b** Body weight (**a**) and percent epiWAT and iBAT weight calculated relative to body weight (**b**) in 15-week-old male WT and *Sirt7* KO mice. $p = 0.0286$ (epiWAT), $p = 0.0288$ (iBAT). **c–j** Data of indirect calorimetry experiments from 15-week-old male WT and *Sirt7* KO mice. $VO_2$ rates (**c**), average $VO_2$ (**d**), average $VCO_2$ (**e**), energy expenditure (**f**), RER (**g**), locomotor activity (**h**), water intake (**i**), and food intake (**j**). $p = 0.0021$ in (**c**); $p = 0.0023$ (Dark), $p = 0.0082$ (Light), $p = 0.0021$ (Total) in (**d**); $p = 0.0255$ (Dark), $p = 0.0101$ (Light), $p = 0.0106$ (Total) in (**e**); $p = 0.0063$ (Dark), $p = 0.0049$ (Light), $p = 0.0026$ (Total) in (**f**);

$p = 0.0296$ in (**j**). **k** Representative H&E-stained sections of iBAT from 15-week-old male WT and *Sirt7* KO mice (left panel, scale bar = 50 μm), and quantification of the lipid area ($n = 3$ independent animals per group) (right panel). $p = 0.0347$. **l** Oscillation of body temperature in 10-week-old male WT and *Sirt7* KO mice. $p = 0.0013$. Data are presented as means ± SEM. All numbers ($n$) are biologically independent samples. Two-way ANOVA with Bonferroni's multiple comparisons test (**c**, **l**); two-tailed Student's *t*-test (**a**, **b**, **d–k**). *$p < 0.05$; **$p < 0.01$. Source data are provided as a Source Data file.

the data showed a typical daily torpor pattern with a severe drop in energy expenditure after 20 h of fasting and a significant drop in body temperature after 24 h (Supplementary Fig. 2g, h). Under these conditions, both 10-week-old WT and *Sirt7* KO mice entered daily torpor, as indicated by energy expenditure. Nevertheless, *Sirt7* KO mice constantly consumed more energy than WT mice (Fig. 2d, e). The body temperature of *Sirt7* KO mice was significantly higher than that of WT mice, which remained low (Fig. 2f). These results indicate that SIRT7 is a fundamental factor involved in energy saving, even in the hypometabolic state, but not a causative factor of suppressed energy expenditure and body temperature in daily torpor.

## SIRT7 suppresses energy expenditure and thermogenesis via multiple pathways

To elucidate the mechanism of SIRT7 regulation in BAT thermogenesis, we first analyzed the expression of BAT-related genes in iBAT of 15-week-old WT and *Sirt7* KO mice. The expression levels of *deiodinase, iodothyronine, type II* (*Dio2*) and *Ucp1*, which are involved in BAT thermogenesis[16,17,43], were significantly increased in *Sirt7* KO mice (Fig. 3a). In contrast, the expression levels of genes related to brown

adipocyte differentiation were unchanged (*peroxisome proliferative activated receptor, gamma* (*Pparg*), *Prdm16*, *Ppargc1a*) or altered very slightly (*early B cell factor 2* (*Ebf2*)) (Fig. 3a). The *Dio2* gene encodes a type 2 deiodinase that activates thyroid hormone by converting the prohormone thyroxine (T4) to bioactive triiodothyronine (T3). As expected, the T3 level was increased in iBAT of *Sirt7* KO mice, but not in serum (Fig. 3b). The protein level of UCP1 in iBAT was markedly higher in *Sirt7* KO mice than in WT mice, even after taking into consideration the differences in mRNA expression (Fig. 3c). The expression of *ELOVL fatty acid elongase 3* (*Elovl3*), whose expression is enhanced in response to cold-induced sympathetic nerve stimulation, was also elevated (Fig. 3a). In inguinal WAT, the expression of *Ucp1*, *Dio2*, and *Elovl3* tended to increase in *Sirt7* KO mice, suggesting that SIRT7 deficiency enhances the browning signature of WAT (Supplementary Fig. 3a). Furthermore, the expression levels of genes promoting functional sympathetic innervation in BAT (*Clstn3b* and *S100b*)[44] and remodeling of the neurovascular networks in BAT and WAT (*bone morphogenetic protein 8b* (*Bmp8b*))[45] were significantly elevated in *Sirt7* KO mice (Fig. 3a). Consistent with this finding, neuronal tyrosine hydroxylase (TH) content in iBAT was markedly increased in *Sirt7* KO

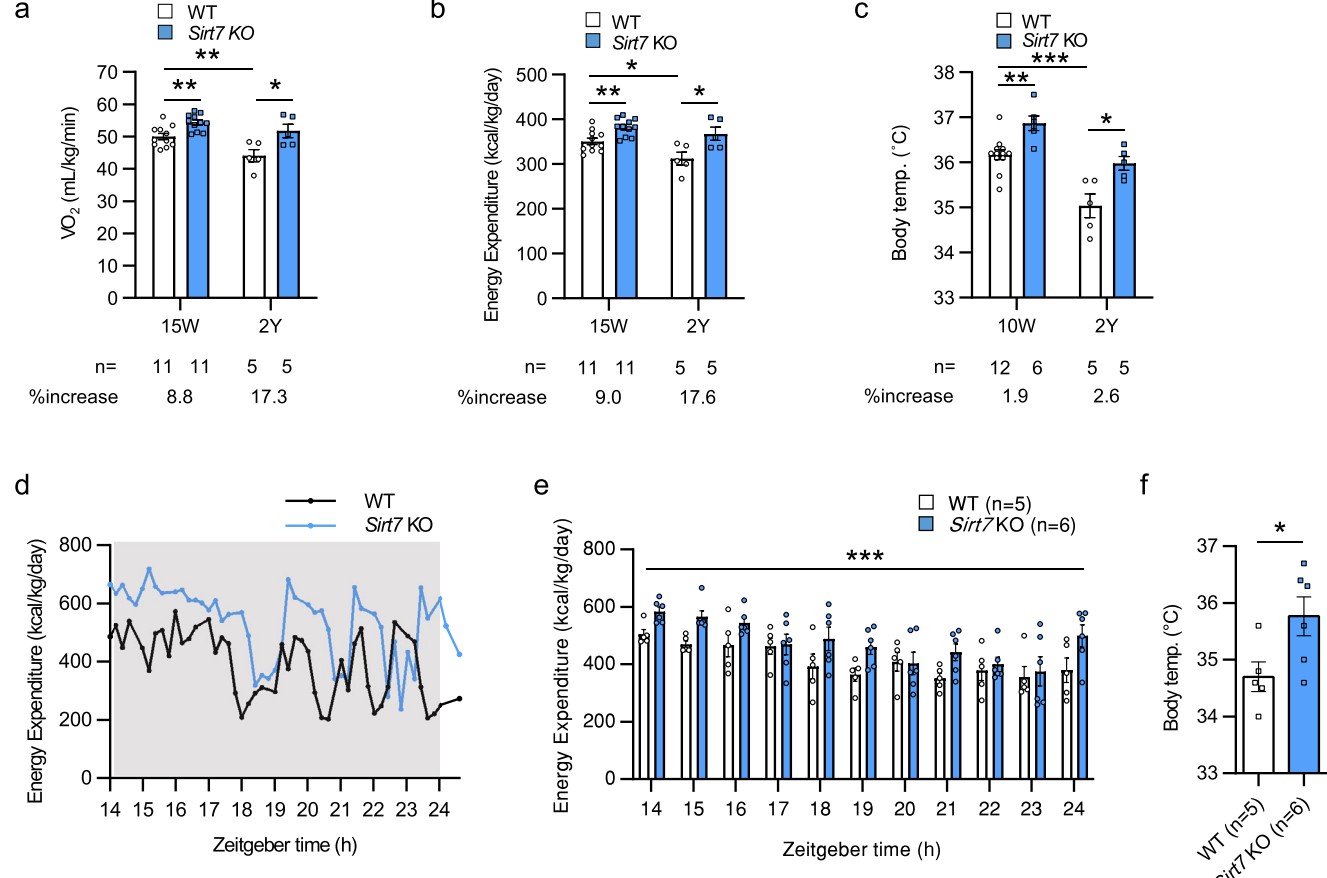

**Fig. 2 | *Sirt7* KO mice exhibit excessive energy expenditure and thermogenesis in the hypometabolic state. a–c** Data of indirect calorimetry experiments and body temperature from 2-year-old male WT and *Sirt7* KO mice (n = 5 independent animals per group). Data from young mice (from Fig. 1d, f, l) are shown for a comparison of their elevated rate with that of the 2-year-old mice. Average VO$_2$ (**a**), energy expenditure (**b**), and body temperature at zeitgeber time (ZT) 10 (**c**). $p = 0.0021$ (15 weeks), $p = 0.0260$ (2 years), $p = 0.0087$ (WT 15 weeks vs. 2 years) in (**a**); $p = 0.0026$ (15 weeks), $p = 0.0283$ (2 years), $p = 0.0170$ (WT 15 weeks vs. 2 years) in (**b**); $p = 0.0025$ (15 weeks), $p = 0.0143$ (2 years), $p = 0.0002$ (WT 15 weeks vs. 2 years) in (**c**). **d–f** Data of indirect calorimetry experiments and body temperature during artificially-induced daily torpor in 10-week-old male WT and *Sirt7* KO mice. Representative pattern of energy expenditure (**d**) and the average energy expenditure (**e**) from 12 h (ZT14) after initiation of fasting (ZT2). Body temperature after 24 h fasting (**f**). $p = 1.4E–05$ in (**e**); $p = 0.0406$ in (**f**). Data are presented as means ± SEM. All numbers (n) are biologically independent samples. Two-way ANOVA with Bonferroni's multiple comparisons test (**e**); two-tailed Student's *t*-test (**a–d, f**). *$p < 0.05$; **$p < 0.01$. Source data are provided as a Source Data file.

mice (Fig. 3d). These data indicate that SIRT7 suppresses BAT thermogenesis via several pathways, including UCP1, thyroid hormone, and the sympathetic nervous system.

Aged *Sirt7* KO mice exhibit hepatic microvesicular steatosis due to mitochondrial dysfunction[37]. In contrast, SIRT7 is reported to suppress mitochondrial function in young hematopoietic stem cells[46]. These findings suggest that SIRT7 exerts divergent functions in mitochondria in different cells. Therefore, we examined the mitochondrial functions in iBAT of *Sirt7* KO mice. The protein levels of mitochondrial oxidative phosphorylation (OXPHOS) complexes I–IV were equivalent in WT and *Sirt7* KO mice (Fig. 3c). WT and *Sirt7* KO mice also showed no differences in the quantity of mitochondria, which is indicated by the mitochondria DNA (mtDNA) copy number, in iBAT (Supplementary Fig. 3b). Furthermore, the expression of several nuclear-encoded mitochondrial transcripts was reduced in the liver of aged *Sirt7* KO mice, as previously shown[37], whereas we discovered that these genes were not altered in the liver of young *Sirt7* KO mice and in the iBAT of young/aged *Sirt7* KO mice (Supplementary Fig. 3c). These results indicate that SIRT7 is not an important regulator of mitochondrial biogenesis and the mitochondrial unfolded protein response, at least in iBAT.

BAT influences a broad range of systemic metabolisms. The effects of BAT on other organs are largely mediated by BAT-derived

secreted factors termed batokines[10,47]. Therefore, we further analyzed the expression of batokine genes. The expression of *fibroblast growth factor 21* (*Fgf21*) and *neuregulin-4* (*Nrg4*) was significantly increased in *Sirt7* KO mice (Fig. 3e), suggesting that SIRT7 regulates the endocrine function of BAT and thereby acts on other organs such as the brain, liver, and adipose tissue. We next analyzed the liver and the skeletal muscle, which is an important organ for energy metabolism and thermogenesis. Although we could not detect obvious metabolic changes at the mRNA level (Supplementary Fig. 3d, e), it may still be possible that SIRT7 in these organs regulates factors related to energy metabolism at the protein level. The protein levels of OXPHOS complexes I–IV were equivalent in the liver of WT and *Sirt7* KO mice, but complex I (NDUFB8) and IV (MTCO1) were decreased in the skeletal muscle of *Sirt7* KO mice (Supplementary Fig. 3f, g). Skeletal muscle might not contribute to the enhanced whole-body energy expenditure in *Sirt7* KO mice. To further explore the role of SIRT7 in energy expenditure and thermogenesis by non-BAT organs, we housed *Sirt7* KO mice for 4 days in a thermoneutral environment (30 °C) in which sympathetic stimulation and other external cues are minimal. As shown in Fig. 3f, g, the phenotype of *Sirt7* KO mice, namely, higher VO$_2$ and energy expenditure was remarkably attenuated at thermoneutrality but still significantly higher during the active (dark) phase only. The body temperature of *Sirt7* KO mice was still slightly but

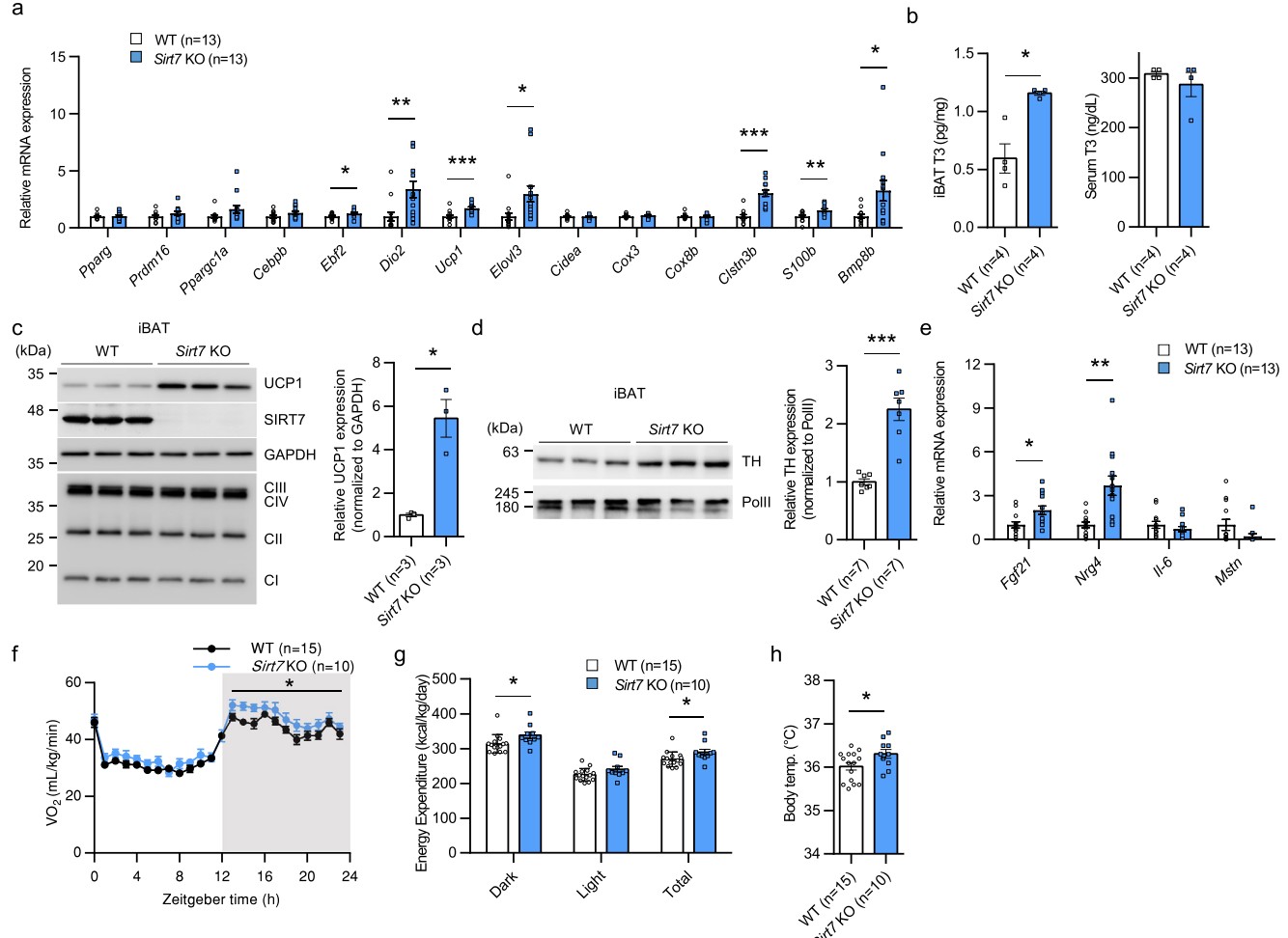

**Fig. 3 | SIRT7 suppresses energy expenditure and thermogenesis via multiple pathways. a** Real-time qPCR analysis of BAT-related genes in iBAT of 15-week-old male WT and *Sirt7* KO mice. $p = 0.0288$ (*Ebf2*), $p = 0.0098$ (*Dio2*), $p = 0.0003$ (*Ucp1*), $p = 0.0206$ (*Elovl3*), $p = 3.4E–06$ (*Clstn3b*), $p = 0.0053$ (*S100b*), $p = 0.0296$ (*Bmp8b*). **b** T3 level in iBAT and serum of 13-week-old male WT and *Sirt7* KO mice. $p = 0.0192$. **c** Western blot analysis of UCP1 and OXPHOS complexes I–IV in iBAT of 15-week-old male WT and *Sirt7* KO mice (left panel), and quantification of the UCP1 protein bands relative to GAPDH control (right panel). $p = 0.0347$. **d** Western blot analysis of tyrosine hydroxylase (TH) in iBAT of 15-week-old male WT and *Sirt7* KO mice (left panel), and quantification of the TH protein bands relative to RNA Polymerase II

(Pol II) control (right panel). $p = 0.0005$. **e** Real-time qPCR analysis of batokine genes in iBAT of 15-week-old male WT and *Sirt7* KO mice. $p = 0.0109$ (*Fgf21*), $p = 0.0016$ (*Nrg4*). **f–h** Data of indirect calorimetry experiments and body temperature from 20-week-old male WT and *Sirt7* KO mice at thermoneutrality. VO$_2$ rates (**f**), energy expenditure (**g**), and body temperature at ZT10 (**h**). $p = 4.5E–02$ (Dark) in (**f**); $p = 0.0386$ (Dark), $p = 0.0473$ (Total) in (**g**); $p = 0.0439$ in (**h**). Data are presented as means ± SEM. All numbers (n) are biologically independent samples. Two-way ANOVA with Bonferroni's multiple comparisons test (**f**); two-tailed Student's $t$-test (**a–e**, **g–h**). *$p < 0.05$, **$p < 0.01$, ***$p < 0.001$. Source data are provided as a Source Data file.

significantly higher than that of WT mice (Fig. 3h). In this condition, there was no difference in the expression of *Fgf21 and Nrg4* between WT and *Sirt7* KO mice (Supplementary Fig. 3h). Taken together, these results indicate that SIRT7 suppresses energy expenditure and thermogenesis via multiple pathways, largely in BAT, but also in other organs.

**Brown adipocytic SIRT7 suppresses energy expenditure and thermogenesis in vivo**

To define the intrinsic role of SIRT7 in brown adipocytes in vivo, we generated two lines of conditional *Sirt7* KO mice: adipose tissue-specific *Sirt7* KO (*Sirt7* AdKO) mice using *Adipoq-Cre* mice[48] and BAT-specific *Sirt7* KO (*Sirt7* BAdKO) mice using *Ucp1-Cre* mice[49].

In 12-week-old male *Sirt7* AdKO mice, the body weight was nearly the same as that of control mice, although iBAT weighed less (Supplementary Fig. 4a, b). VO$_2$, VCO$_2$, energy expenditure, and body temperature were significantly increased in *Sirt7* AdKO mice (Fig. 4a–c and Supplementary Fig. 4c, d), whereas the RER, locomotor activity, and water intake were unchanged (Supplementary Fig. 4e–g) and food

intake versus body weight was slightly increased (Supplementary Fig. 4h). These results demonstrate that adipocytic SIRT7 is important for the suppression of energy expenditure and thermogenesis in vivo. Increased expression of genes in *Sirt7* KO mice, such as *Dio2*, *Fgf21*, and *Nrg4*, was confirmed in *Sirt7* AdKO mice, whereas *Ucp1* and *Elovl3* expression was unchanged (Fig. 4d). In accordance with this change, the T3 level was increased in iBAT of *Sirt7* AdKO mice (Fig. 4e). Interestingly, the protein level of UCP1 was significantly higher in iBAT of *Sirt7* AdKO mice, without change in the mRNA level (Fig. 4f). In inguinal WAT, the expression of *Ppargc1a* and *Elovl3* tended to be high in *Sirt7* AdKO mice (Supplementary Fig. 4i). Because the browning phenotype is not clear at room temperature (23 °C), mice were administered 1 mg/kg/day norepinephrine for 5 days to induce WAT browning. The results clearly showed that SIRT7 deficiency enhances the browning of inguinal WAT (Supplementary Fig. 4j).

Furthermore, the phenotypes of 12-week-old male *Sirt7* BAdKO mice regarding energy expenditure and thermogenesis were almost identical to those of the *Sirt7* KO and *Sirt7* AdKO mice (Fig. 5a–c and Supplementary Fig. 5a–h). These results demonstrate that brown

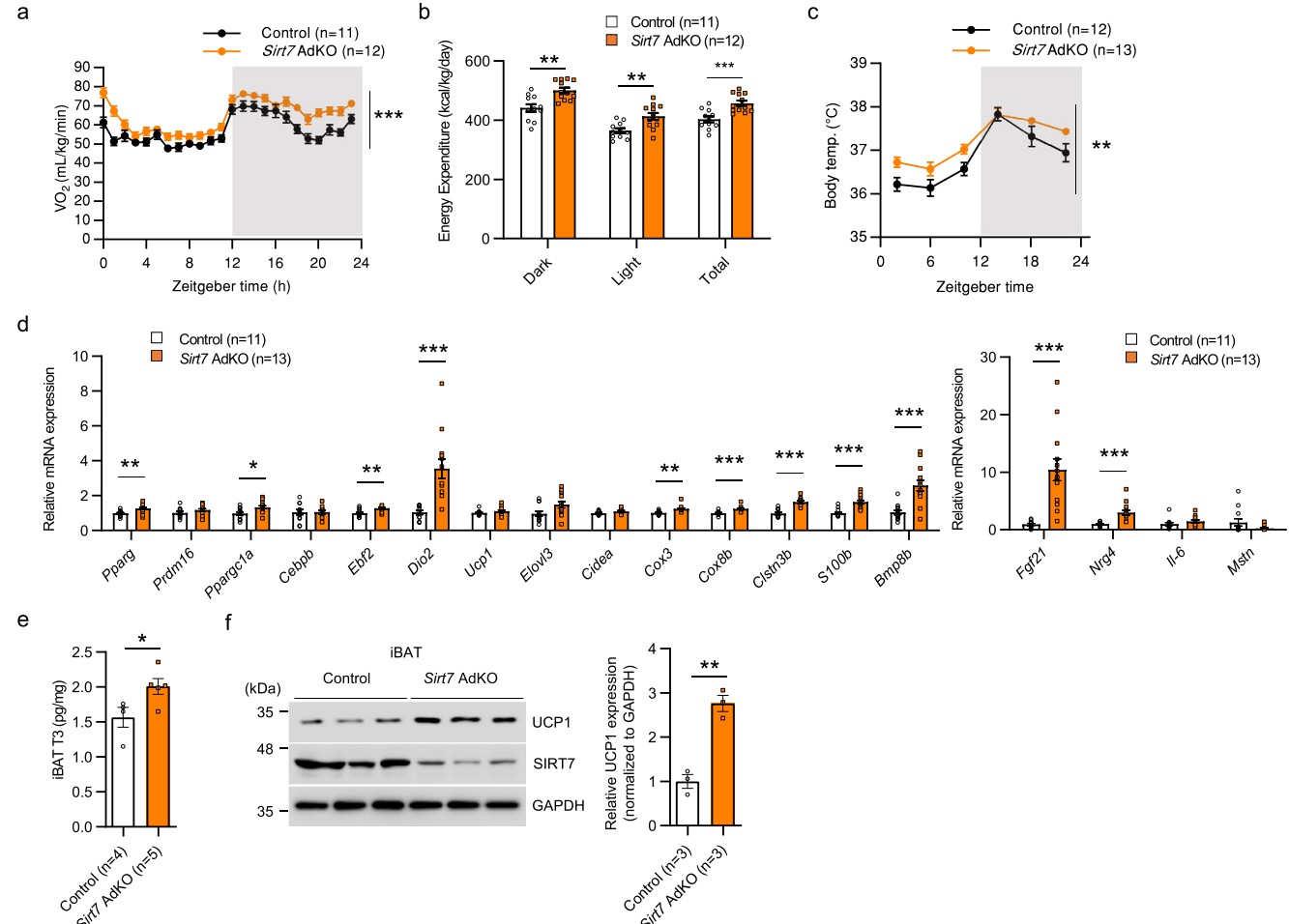

**Fig. 4 | SIRT7 deficiency in adipose tissue elevates whole-body energy expenditure and body temperature. a**, **b** Data of indirect calorimetry experiments and body temperature from 12-week-old male *Adipoq-Cre* control and *Sirt7* AdKO mice. VO$_2$ rates (**a**) and energy expenditure (**b**). $p = 3.7\text{E}{-}29$ in (**a**); $p = 0.0011$ (Dark), $p = 0.0030$ (Light), $p = 0.0006$ (Total) in (**b**). **c** Oscillation of body temperature in 20-week-old male *Adipoq-Cre* control and *Sirt7* AdKO mice. $p = 0.0025$. **d** Real-time qPCR analysis of BAT-related genes in iBAT of 12-week-old male *Adipoq-Cre* control and *Sirt7* AdKO mice. $p = 0.0089$ (*Pparg*), $p = 0.0239$ (*Ppargc1a*), $p = 0.0026$ (*Ebf2*), $p = 0.0005$ (*Dio2*), $p = 0.0010$ (*Cox3*), $p = 0.0006$ (*Cox8b*), $p = 5.0\text{E}{-}06$ (*Clstn3b*),

$p = 0.0001$ (*S100b*), $p = 0.0002$ (*Bmp8b*), $p = 0.0003$ (*Fgf21*), $p = 0.0008$ (*Nrg4*). **e** T3 level in iBAT of 11-week-old male *Adipoq-Cre* control and *Sirt7* AdKO mice. $p = 0.0420$. **f** Western blot analysis of UCP1 in iBAT of 12-week-old male *Adipoq-Cre* control and *Sirt7* AdKO mice (left panel). Quantification of the UCP1 protein bands relative to GAPDH control is shown on the right side. $p = 0.0019$. Data are presented as means ± SEM. All numbers (*n*) are biologically independent samples. Two-way ANOVA with Bonferroni's multiple comparisons test (**a**, **c**); two-tailed Student's *t*-test (**b**, **d**–**f**). *$p < 0.05$, **$p < 0.01$, ***$p < 0.001$. Source data are provided as a Source Data file.

adipocytic SIRT7 suppresses energy expenditure and thermogenesis. The expression of *Fgf21* and *Nrg4* was significantly higher in *Sirt7* BAdKO mice as well, whereas that of other genes that were increased in *Sirt7* AdKO mice, including *Dio2*, was unchanged (Fig. 5d). The T3 level in iBAT was also not increased in *Sirt7* BAdKO mice (Supplementary Fig. 5i). Although *Prdm16* and *CCAAT/enhancer binding protein, beta* (*Cebpb*) expression was unexpectedly increased, it is doubtful that these mRNA expressions reflected the protein levels because their downstream target genes were not activated (Fig. 5d). Notably, there was a significantly higher protein amount of UCP1 in both *Sirt7* AdKO and *Sirt7* BAdKO mice than in control mice (Figs. 5e and 4f), despite an unchanged mRNA level. These findings suggest that brown adipocytic SIRT7 suppresses UCP1 protein expression at the translation or protein stability level. To provide more convincing evidence for the importance of SIRT7 in iBAT, we tested *Sirt7* BAdKO mice at thermoneutrality. The results showed that the phenotype of higher VO$_2$, energy expenditure, and body temperature was almost completely abolished in *Sirt7* BAdKO mice (Fig. 5f, g and Supplementary Fig. 5j). In addition, the expression of *Fgf21 and Nrg4* in iBAT of *Sirt7* BAdKO mice was not highly activated anymore (Supplementary Fig. 5k). Taken

together, our results revealed that brown adipocytic SIRT7 suppresses whole-body energy expenditure and thermogenesis.

**Brown adipocytic SIRT7 attenuates UCP1 protein expression and mitochondrial respiration**

We next verified SIRT7 function using primary cultured brown adipocytes. As shown in Fig. 6a, there was no difference in lipid droplets between WT and *Sirt7* KO mice when the primary stromal vascular fraction (SVF) from iBAT was induced to differentiate into brown adipocytes. None of the BAT-related genes, including genes related to brown adipocyte differentiation, showed altered expression in *Sirt7* KO brown adipocytes (Fig. 6b). These results indicate that SIRT7 deficiency does not influence the differentiation of brown adipocytes. Nevertheless, the protein level of UCP1 was significantly higher in *Sirt7* KO brown adipocytes than in WT cells (Fig. 6c), consistent with our observations in *Sirt7* AdKO and *Sirt7* BAdKO mice (Figs. 4f and 5e). Here, there were no differences in the quantity of mitochondria between WT and *Sirt7* KO brown adipocytes (Fig. 6d). We next determined oxygen consumption rate (OCR) using a Seahorse XF24 Flux Analyzer via the sequential injection of oligomycin, FCCP, and

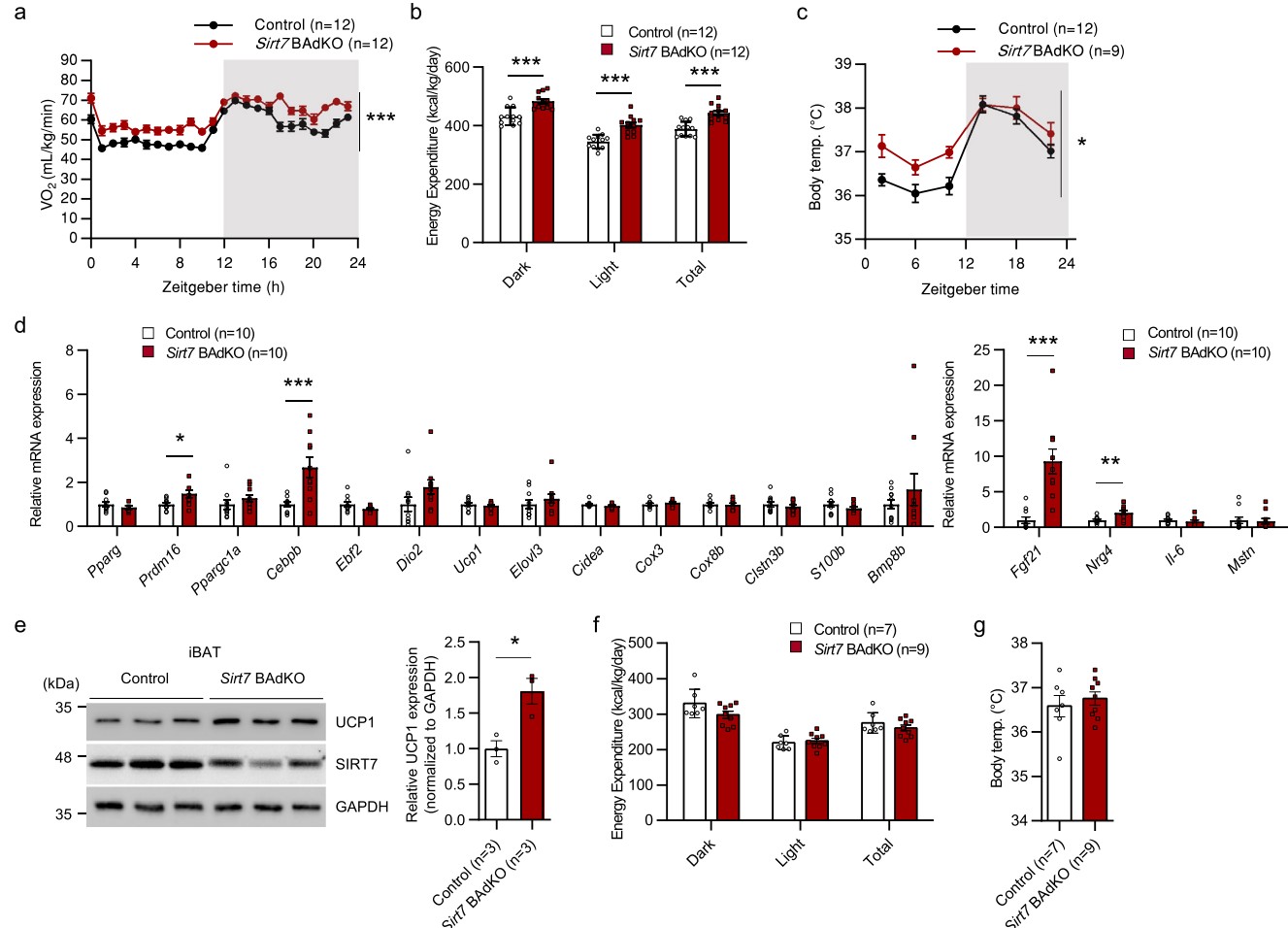

**Fig. 5 | Brown adipocytic SIRT7 suppresses energy expenditure and thermogenesis in vivo. a, b** Data of indirect calorimetry experiments and body temperature from 12-week-old male *Ucp1-Cre* control and *Sirt7* BAdKO mice. VO$_2$ rates (**a**) and energy expenditure (**b**). $p = 9.1E-36$ in (**a**); $p = 1.8E-04$ (Dark), $p = 3.8E-05$ (Light), $p = 3.6E-05$ (Total) in (**b**). **c** Oscillation of body temperature in 20-week-old *Ucp1-Cre* control and *Sirt7* BAdKO mice. $p = 0.0399$. **d** Real-time qPCR analysis of BAT-related genes in iBAT of 12-week-old male *Ucp1-Cre* control and *Sirt7* BAdKO mice. $p = 0.0245$ (*Prdm16*), $p = 0.0046$ (*Cebpb*), $p = 0.0009$ (*Fgf21*), $p = 0.0056$ (*Nrg4*). **e** Western blot analysis of UCP1 in iBAT of 12-week-old male *Ucp1-Cre* control

and *Sirt7* BAdKO mice (left panel). Quantification of the UCP1 protein bands relative to GAPDH control is shown on the right side. $p = 0.0192$. **f, g** Data of indirect calorimetry experiments and body temperature from 20-week-old male *Ucp1-Cre* control and *Sirt7* BAdKO mice at thermoneutrality. Energy expenditure (**f**) and body temperature at ZT10 (**g**). Data are presented as means ± SEM. All numbers (*n*) are biologically independent samples. Two-way ANOVA with Bonferroni's multiple comparisons test (**a, c**); two-tailed Student's *t*-test (**b, d–g**). *$p < 0.05$, **$p < 0.01$, ***$p < 0.001$. Source data are provided as a Source Data file.

antimycin A/Rotenone (Fig. 6e). *Sirt7* KO brown adipocytes displayed a significantly higher basal and maximal respiration of the mitochondria (Fig. 6f). Taken together, these data suggest that SIRT7 suppresses mitochondrial respiration and energy expenditure cell-autonomously via a decrease in the UCP1 protein level in brown adipocytes.

## SIRT7 suppresses energy expenditure via IMP2 in brown adipocytes

Because SIRT7 has been reported to localized to the nucleus and cytosol but not the mitochondria[34], SIRT7 may indirectly attenuate the UCP1 protein level by acting on other factors related to translation or protein stability. Therefore, we systematically searched for molecules interacting with SIRT7 by performing affinity chromatography (Halo-Tag pull-down assay) of protein extract from cultured brown adipocytes, followed by liquid chromatography mass spectrometry/mass spectrometry (LC-MS/MS) analysis (Fig. 7a and Supplementary Data 1). Among the candidate proteins interacting with SIRT7, we detected IMP2, an RNA-binding protein (RBP) that binds the UTR of *Ucp1* mRNA and inhibits its translation[21]. IMP2 also alters the translation of 12 other mRNAs encoding mitochondrial proteins, suggesting the importance of IMP2 in mitochondrial functions[21]. The interaction of SIRT7

and IMP2 was reconfirmed by the pull-down assay and a co-immunoprecipitation assay (Fig. 7b, c). Because *Imp2* KO mice have increased energy expenditure and more UCP1 protein in brown fat[21], we speculated that the activity of IMP2 in brown adipocytes might be promoted by SIRT7. We assessed the contribution of IMP2 protein to *Sirt7* KO brown adipocytes via extracellular flux analysis and western blotting. As shown in Fig. 7d, e, IMP2 disruption using an adeno-associated virus (AAV)-mediated CRISPR/Cas9 system enhanced the basal, maximal, and uncoupled respiration in WT brown adipocytes, whereas the already increased respiration (basal, maximal, and uncoupled) in *Sirt7* KO cells was not further enhanced by IMP2 disruption. The UCP1 protein level was also increased by IMP2 deficiency in WT cells, but not further increased in *Sirt7* KO cells (Fig. 7f), while *Ucp1* mRNA levels were not altered in any samples (Fig. 7g). Thus, these results indicate that SIRT7 suppresses energy expenditure in brown adipocytes, at least partly in an IMP2-dependent manner.

## SIRT7 deacetylates IMP2, thereby attenuating the protein level of UCP1 in BAT

To elucidate the molecular basis of how SIRT7 regulates the function of IMP2, we first examined whether SIRT7 deacetylates IMP2. SIRT7

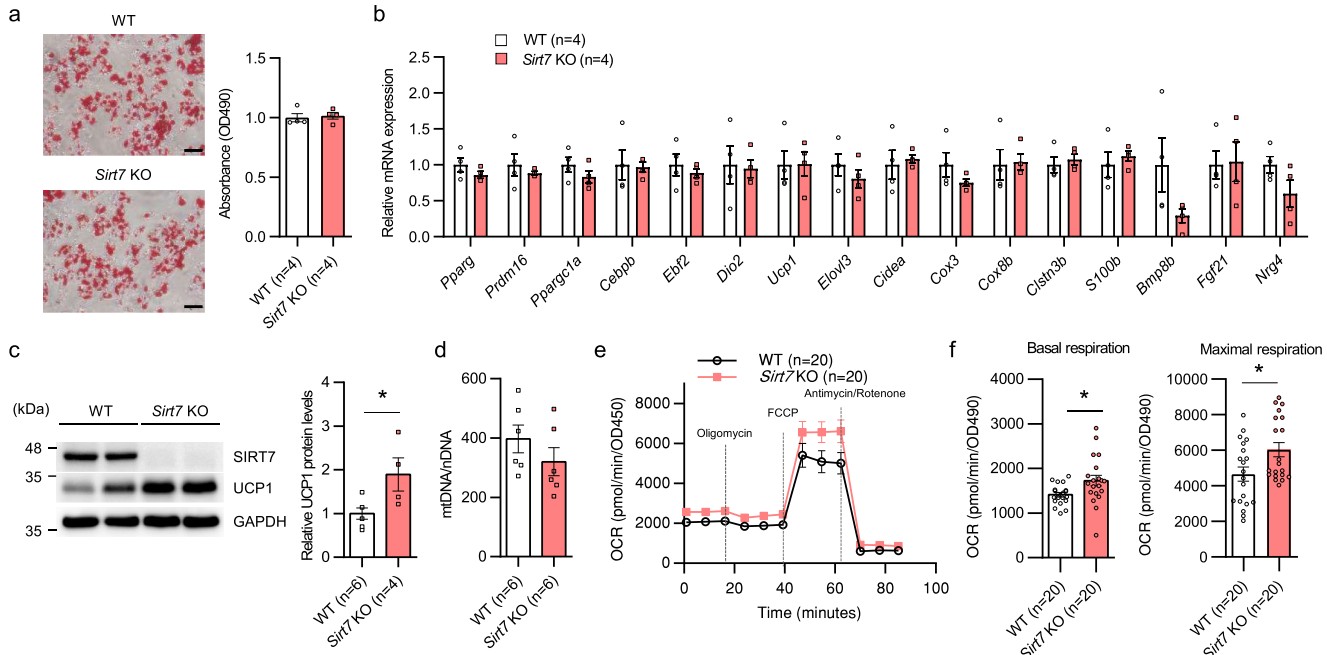

**Fig. 6 | SIRT7 cell-autonomously suppresses mitochondrial respiration via a reduction in the UCP1 protein level in brown adipocytes. a** Oil red O staining of differentiated primary brown adipocytes from WT and *Sirt7* KO mice (left panel, scale bar = 50 μm), and quantification of the stained oil red O (right panel). **b** Real-time qPCR analysis of BAT-related genes in the cells described in (**a**). **c** Western blot analysis of UCP1 in the cells described in (**a**) (left panel), and quantification of the UCP1 protein bands relative to GAPDH control (right panel). *p* = 0.0307.

**d** Evaluation of the mtDNA copy number in differentiated primary brown adipocytes from WT and *Sirt7* KO mice by determining the ratio of mtDNA to nDNA. **e, f** Mitochondrial stress test in the cells described in (**a**). Time course OCR (**e**) and quantification of mitochondrial respiration (**f**). *p* = 0.0247 (basal), *p* = 0.0239 (maximal). Data are presented as means ± SEM. All numbers (*n*) are biologically independent samples. **p* < 0.05 by two-tailed Student's *t*-test. Source data are provided as a Source Data file.

reduced the acetylation of IMP2 in HEK293T cells expressing p300 acetyl transferase, whereas SIRT7[H188Y] (a loss-of-function mutant) had no effect on IMP2 acetylation (Fig. 8a). Furthermore, the acetylation of endogenous IMP2 was enhanced in iBAT from *Sirt7* KO mice (Fig. 8b). These results indicate that SIRT7 deacetylates IMP2 in brown fat.

To map the region of IMP2 involved in its interaction with SIRT7, a Halo-SIRT7 pull-down assay was performed with lysates of HEK293T cells expressing IMP2 deletion mutants fused to the GAL4 DNA-binding domain (GAL4DBD). As shown in Fig. 8c, the C-terminal region [381–592] of IMP2 bound to SIRT7. Further studies with additional deletion mutants revealed that SIRT7 specifically bound to the C1A region [381–481] without part of C1B [442–494] (Fig. 8c and Supplementary Fig. 6a). To identify the residues targeted by SIRT7, we introduced a deacetylation-mimicking lysine (K)-to-arginine (R) mutation into each of the two residues (K438 and K439) located in this region [381–441] and assessed the effect of SIRT7-mediated deacetylation. Although SIRT7 deacetylated IMP2[K439R], it did not further deacetylate IMP2[K438R], indicating that K438 is targeted for deacetylation by SIRT7 (Fig. 8d and Supplementary Fig. 6b).

K438 is located at the highly conserved GxxG loop within KH domain 3, which is critical for binding to target RNAs[50,51] (Supplementary Fig. 6c). Thus, we suspected that the acetylation status of K438 of IMP2 would impact the translational efficiency of *Ucp1* mRNA. Previous work reported that alternative polyadenylation creates *Ucp1* transcripts carrying long or short 3'-UTRs in mice and that the long form is important for producing UCP1 for thermogenesis[23]. First, we verified by RNA immunoprecipitation assay using in vitro-transcribed RNA that IMP2 naturally binds to long form, but not to the short form (Supplementary Fig. 6d). Therefore, we analyzed the binding of IMP2[WT], IMP2[K438R], and IMP2[K438Q] (acetylation-mimicking mutant) to long *Ucp1* 3'-UTR. The binding of IMP2[K438Q] to *Ucp1* 3'-UTR was significantly low compared with IMP2[WT] and IMP2[K438R] (Fig. 8e). Recombinant IMP2 protein can inhibits in vitro translation from *luciferase-*

*Ucp1* UTR fusion mRNA[21]. Therefore, we generated recombinant protein in *E. coli* for IMP2[WT], IMP2[K438R], and IMP2[K438Q] (Supplementary Fig. 6e) and performed an in vitro translation-based luciferase assay. As expected, recombinant IMP2[WT] and IMP2[K438R] protein progressively inhibited *luciferase-Ucp1* UTR mRNA translation (Fig. 8f). In contrast, this inhibitory function was severely impaired in IMP2[K438Q] (Fig. 8f). These results support the hypothesis that SIRT7-dependent deacetylation of IMP2 leads to attenuation of the steady-state UCP1 protein level in BAT.

## Discussion

Both suppression and activation of BAT thermogenesis are critical for maintaining energy homeostasis. Although several sirtuins positively participate in the regulation of BAT functions, we revealed that SIRT7 clearly plays the opposite role in BAT thermogenesis. Our results demonstrated that SIRT7 deficiency in brown adipocytes leads to higher energy expenditure both in vitro and in vivo. Crucially, different *Sirt7* KO mouse lines were incapable of properly decreasing their body temperature and energy expenditure, even in a reduced metabolic state such as aging or daily torpor. We also revealed that SIRT7 suppressed brown fat thermogenesis by attenuating *Ucp1* mRNA translation via deacetylation of IMP2 (Fig. 8g). Thus, this study identified SIRT7 as the only energy-saving factor in the sirtuin family capable of suppressing BAT thermogenesis. In addition, although it is recognized that SIRT1, −3, −5, and −6 are positive regulators of BAT thermogenesis, SIRT1, −3, and −5 do not affect the thermogenic function of BAT without a stressor such as HFD or cold challenge[29,31,32,52]. Taken together, the results show that SIRT6 and SIRT7 acting antagonistically are critical regulators of BAT thermogenesis under normal conditions.

We demonstrated here that acetylation of IMP2 at K438, located in the GxxG loop within KH domain 3, impairs the ability of IMP2 to inhibit the translation of *Ucp1* mRNA in BAT and that SIRT7

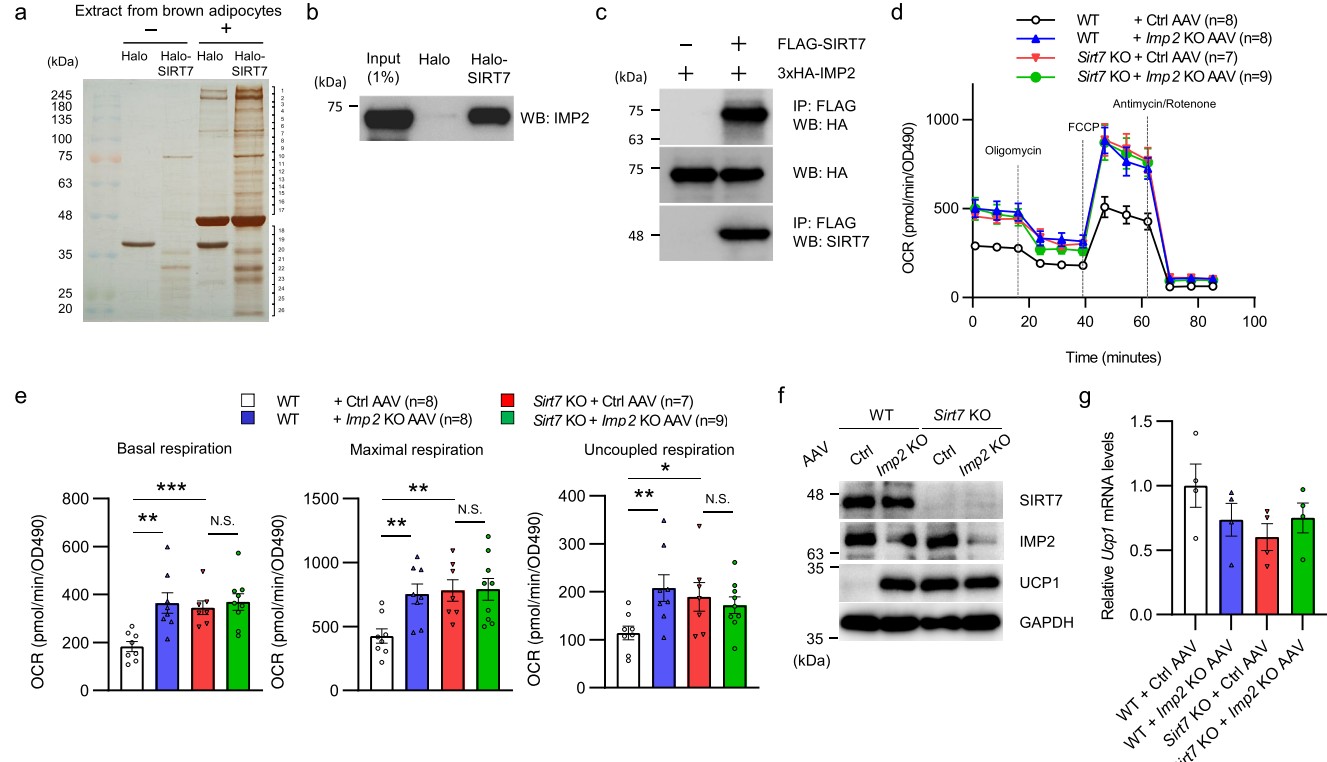

**Fig. 7 | SIRT7 suppresses energy expenditure via IMP2 in brown adipocytes.**
**a** Halo-SIRT7 pull-down assay with extracts from brown adipocytes. Eluted proteins were resolved by SDS-PAGE, followed by silver staining. The gel lanes were cut into 26 pieces and the proteins within each gel piece were analyzed by mass spectrometry. **b** Halo-SIRT7 pull-down assay performed with lysates from HEK293T cells overexpressing 3×HA-IMP2. Overexpressed IMP2 was detected by WB with an anti-IMP2 antibody. **c** Co-IP assay detecting the interaction between FLAG-SIRT7 and 3×HA-IMP2 in HEK293T cells. **d**, **e** The effect of *Imp2* deficiency on mitochondrial respiration in differentiated primary brown adipocytes from WT and *Sirt7* KO mice. SVF cells were infected with the indicated recombinant AAV and differentiated for 9 days. Time course OCR (**d**) and quantification of mitochondrial respiration (**e**).

$p = 0.0020$ (WT + Ctrl AAV vs. WT + *Imp2* KO AAV), $p = 0.0005$ (WT + Ctrl AAV vs. *Sirt7* KO + Ctrl AAV) in basal respiration; $p = 0.0039$ (WT + Ctrl AAV vs. WT + *Imp2* KO AAV), $p = 0.0031$ (WT + Ctrl AAV vs. *Sirt7* KO + Ctrl AAV) in maximal respiration; $p = 0.0099$ (WT + Ctrl AAV vs. WT + *Imp2* KO AAV), $p = 0.0348$ (WT + Ctrl AAV vs. *Sirt7* KO + Ctrl AAV) in uncoupled respiration. **f**, **g** Western blot (**f**) and real-time qPCR (**g**) analysis of Ucp1 in the cells described in (**d**). $n = 4$ independent samples per group in (**g**). WB western blotting, IP immunoprecipitation, N.S. not significant. Data are presented as means ± SEM. All numbers (*n*) are biologically independent samples. The screening experiment (**a**) were performed one time. *$p < 0.05$, **$p < 0.01$ by two-tailed Student's *t*-test. Source data are provided as a Source Data file.

deacetylates this position. It is well established that the phosphorylation of RBPs is critical for their interaction with RNA and their regulation of RNA processing[51] but, to our knowledge, there is no study to report that regulation of acetylation in the GxxG loop of the KH domain-containing RBP family plays a critical role in their functions. Of the 39 KH domain family members, 10 have been reported as candidate SIRT7-interacting proteins[53] (Supplementary Fig. 6f), suggesting the existence of a common mechanism for the regulation of RBP function via acetylation of the KH domain. The mammalian IMP family (IMP1–3) plays an important role in development, tumorigenesis, and stemness through the regulation of various aspects of RNA processing, such as stability, translation, and localization[54,55]. Recent studies have highlighted the roles of IMP2 in the metabolic control of multiple organs, including BAT, muscle, liver, and pancreatic islets, via the post-transcriptional regulation of different genes[56]. In addition, genome-wide association studies have revealed that a cluster of single nucleotide polymorphisms in the second intron of human *IMP2* is implicated in type 2 diabetes[56]. Therefore, it is highly likely that SIRT7-dependent deacetylation of IMP2 at K438 affects the processing of the other target mRNAs in multiple organs. Because IMP2 has two RNA-recognition motifs (RRMs) followed by four KH domains for the proper binding of individual RNAs[50], acetylation of IMP2 at K438 may affect only some of the whole target mRNAs. Further study is required to determine the transcripts modulated by acetylation of IMP2 at K438 and whether SIRT7 deacetylates IMP2 in other organs as well.

Furthermore, in addition to the IMP2-UCP1 pathway, we presented the possibility that SIRT7 suppresses BAT thermogenesis by attenuating the production of T3 and the sympathetic innervation in BAT. The expression of *Dio2* and *Clstn3b* was significantly increased in iBAT of *Sirt7* KO mice and *Sirt7* AdKO mice, but not in that of *Sirt7* BAdKO mice. How does SIRT7 regulate the expression of these genes? Recent studies identified A-FABP, FGF6, and FGF9 as adipokines regulating the expression of BAT-related genes, such as *Dio2*[57,58]. Thus, white adipocytic SIRT7 may regulate the secretion of such adipokines and indirectly regulate *Dio2* and *Clstn3b* expression in iBAT. In addition, the possibility remains that the signaling pathway promoting *Dio2* expression by extracellular factors is enhanced in *Sirt7* KO brown adipocytes, because a nonsignificant tendency was also observed for increased *Dio2* expression in iBAT of *Sirt7* BAdKO mice. Similarly, *Ucp1* expression was significantly increased in *Sirt7* KO mice, but not in the other models. *Ucp1* gene expression is highly induced by sympathetic stimulation, and the expression levels of *Ucp1* in iBAT of *Sirt7* KO mice at thermoneutrality was no longer high (Supplementary Fig. 3h), suggesting that brain SIRT7 may control *Ucp1* expression in iBAT via the sympathetic nervous system. Moreover, it is probable that a variation in the depletion efficiency of SIRT7 by different Cre drivers (Figs. 4f, 5e) affected these observed changes in gene expression.

BAT regulates whole-body energy expenditure through not only BAT thermogenesis but also endocrine action. Given that *Sirt7* KO, *Sirt7* AdKO, and *Sirt7* BAdKO mice exhibit a significant increase in

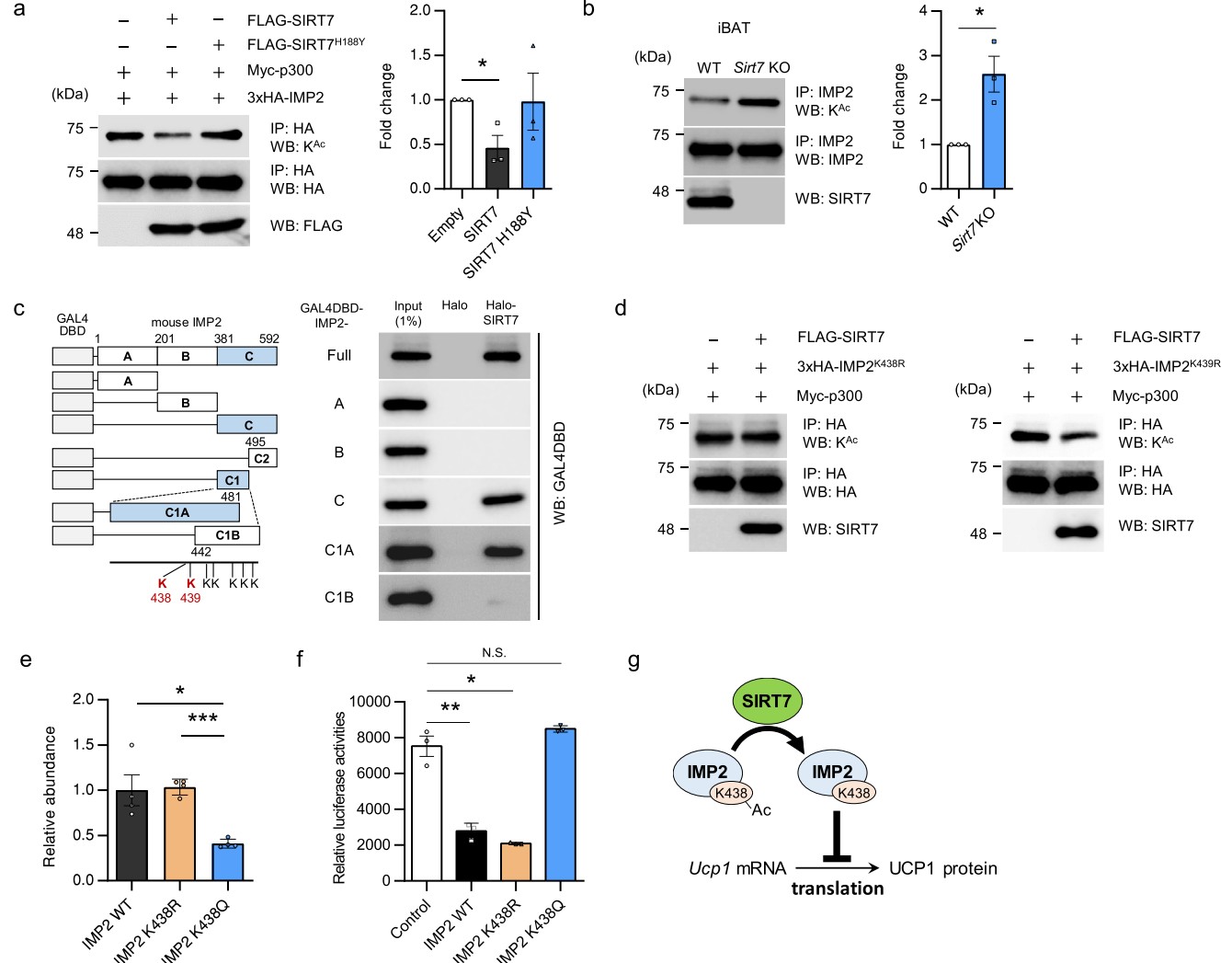

**Fig. 8 | Deacetylation of IMP2 by SIRT7 attenuates the translation of *Ucp1* mRNA. a** The effect of SIRT7 overexpression on IMP2 acetylation. HEK293T cells were transfected with the indicated expression plasmid, and IMP2 acetylation was assessed by IP and WB (left panel). Quantification of the acetylated IMP2 relative to IMP2 ($n = 3$) (right panel). $p = 0.0174$. **b** The effect of *Sirt7* deficiency on IMP2 acetylation. Protein lysates of iBAT were subjected to IP, after which the acetylated IMP2 was detected by WB (left panel). Quantification of the acetylated IMP2 relative to IMP2 ($n = 3$) (right panel). $p = 0.0170$. **c** Mapping of the region of the interaction between IMP2 and SIRT7. Halo-SIRT7 pull-down assay with lysates of HEK293T cells expressing the indicated IMP2 deletion mutants fused with GAL4DBD. See also Supplementary Fig. 6a. **d** The effect of SIRT7 overexpression on IMP2$^{K438R}$ and IMP2$^{K439R}$ acetylation. HEK293T cells were transfected with the indicated expression plasmid, and IMP2 acetylation was assessed by IP and WB. Quantification of the acetylated IMP2 is shown in Supplementary Fig. 6b. **e** Binding between in vitro-

transcribed long *Ucp1* 3'-UTR and IMP2 (IMP2$^{WT}$, IMP2$^{K438R}$, IMP2$^{K438Q}$) ($n = 4$ independent samples per group). $p = 0.0401$ (IMP2$^{WT}$ vs. IMP2$^{K438Q}$), $p = 1.7E-05$ (IMP2$^{K438R}$ vs. IMP2$^{K438R}$). **f** The effect of IMP2 mutant on the translation of *luciferase-Ucp1* UTR fusion mRNA. *luciferase-Ucp1* UTR mRNA was transcribed in vitro and used in an insect cell-free translation system with recombinant protein in *E. coli* for IMP2$^{WT}$, IMP2$^{K438R}$, and IMP2$^{K438Q}$ ($n = 3$ independent samples per group). Translation was estimated by the gain in luciferase activity after incubation for 5 h at 25 °C. $p = 0.0016$ (Control vs. IMP2$^{WT}$), $p = 0.0106$ (Control vs. IMP2$^{K438R}$). **g** Proposed model for the suppression of brown fat thermogenesis through attenuation of *Ucp1* mRNA translation via SIRT7-dependent deacetylation of IMP2. See Discussion for details. WB western blotting, IP immunoprecipitation, N.S. not significant. Data are presented as means ± SEM. All numbers ($n$) are biologically independent samples. *$p < 0.05$, **$p < 0.01$ by two-tailed Student's *t*-test. Source data are provided as a Source Data file.

energy expenditure, it is likely that other major organs are activated by brown adipocytic SIRT7. Indeed, the expression of *Fgf21 and Nrg4* in iBAT was highly elevated in KO types of all three lines. FGF21 is an important regulator of insulin sensitivity, systemic metabolism, and energy expenditure[47]. Although FGF21 is generated primarily in the liver, it is also secreted from BAT and contributes to increased circulating levels only when the organ is highly activated such as during cold exposure[59]. NRG4 attenuates hepatic lipogenesis, augments fuel oxidation and energy expenditure, and maintains a healthy adipokine profile[60]. Unfortunately, it was hard to evaluate the contribution of elevated *Fgf21* and *Nrg4* expression to whole-body energy expenditure because these elevated expressions were not seen at thermoneutrality (Supplementary Fig. 5k). A different approach, such as double-KO mice

or a neutralizing antibody, would help us to assess the impact of SIRT7-dependent batokine expression.

Because our study was conducted only in mouse models (although many different mouse models were used), it remains to be determined whether our findings can directly translate to humans. In adult humans, BAT is present in multiple locations, and UCP1-positive adipocytes from the supraclavicular region show a molecular signature resembling that of mouse beige adipocytes, whereas the deep neck region contains classical brown adipocyte-like cells. As mentioned above, previous studies have emphasized the importance of sirtuins in the browning of WAT rather than in the classical BAT. For example, SIRT1 gain-of-function induces browning of WAT in vivo by deacetylating PPARγ at K293 and K268, whereas no effects of SIRT1 gain- or

loss-of-function were detected in iBAT[52]. In the present study, we focused mainly on classical brown adipocytes, but we are also confident that SIRT7 plays an important role in the browning of WAT. That is because, first, we observed that the expression of browning-related genes was clearly higher in inguinal WAT of *Sirt7* AdKO mice treated with norepinephrine than that of control mice (Supplementary Fig. 4j). Second, we recently found that SIRT7 deacetylates PPARγ2 at K382 and enhances fat accumulation in white adipocytes by regulating the expression of genes involved in lipogenesis[61]. In a review of the RNA-seq data of this report, gene ontology analysis of the upregulated genes in PPARγ2$^{K382Q}$-expressing adipocytes showed significant enrichment in adipose tissue development (GO:0060612; $p = 1.1 \times 10^{-3}$) and brown fat cell differentiation (GO:0050873; $p = 6.6 \times 10^{-3}$). Because sirtuin regulates various targets in a cell-dependent manner, its roles vary in different organs. Accordingly, further studies are warranted to uncover the role and precise mechanism of SIRT7 in adipose tissue browning.

It is generally accepted that BAT activation and/or WAT browning have various health benefits. This theory is specifically applicable to individuals consuming excessive nutrients, particularly those with obesity or metabolic syndrome, but not to all other people. Recent studies have identified that BAT activation and/or WAT browning contribute to the development and progression of hypermetabolism in other pathological conditions, such as cachexia associated with cancer, burn injuries, and infectious diseases[62,63]. In addition, harmful adverse effects of browning agents have been indicated. Therefore, we expect the development of innovative strategies that tightly regulate BAT activation and/or WAT browning by using a brake-switch system. SIRT7, as a brake-switch of energy consumption, will hopefully be a potential target for the treatment of metabolic disorders.

## Methods

### Mouse models

All experimental procedures were approved by the Kumamoto University Ethics Review Committee for Animal Experimentation (Approval ID: A27-024, A29-001, A 2019-048, A 2021-001). All mice were housed at a maximum of 5 mice/cage (CL-0133 Plastic cage, CLEA Japan: Dimensions/169 × 376 × 145 mm, Floor area/635 cm², Body materials/TPX, Lid materials/Stainless steel). As bedding, aspen chips (CL-4169, CLEA Japan) were provided. And all mice were maintained in a climate-controlled environment at approximately 22–23 °C and 40–80% humidity under specific pathogen-free conditions and strict 12-h light/dark cycles and had access to regular chow (CE-2, CLEA Japan Inc.) and water *ad libitum*, unless otherwise specified. Both the research team and the veterinary staff monitored health of mice daily by weight, food and water intake, and general assessment of animal activity, panting, and fur condition. Mice showing signs of morbidity (immobility, lack of responsiveness to manual stimulation, and inability to eat or drink) were euthanized by manual cervical dislocation according to the institutional animal care guidelines of Kumamoto University. When mice reach an experimental endpoint, all mice were euthanized using overdose of isoflurane anesthesia. Our *Sirt7* KO mice[64] and another independent line of *Sirt7* KO mice[37] were back-crossed for over five generations with C57BL/6 J (CLEA Japan). Only the heterozygotes were bred (*Sirt7*$^{+/-}$ × *Sirt7*$^{+/-}$), and the littermates (WT and *Sirt7* KO mice) were used for studies. These two lines of *Sirt7* KO mice did not exhibit a phenotype with partial embryonic lethality, postnatal death, growth retardation, and a progeroid-like features, as shown by another *Sirt7* KO mouse strain (JAX012771). *Adipoq-Cre* (JAX010803)[48] and *Ucp1-Cre* (JAX024670)[49] mice, which were back-crossed for over eight generations with C57BL/6 J, were a kind gift from Dr. Evan D. Rosen and were crossed with *Sirt7* floxed mice (*Sirt7*$^{fl/fl}$)[35] for generation of *Sirt7* AdKO mice and *Sirt7* BAdKO mice, respectively.

### Cell culture

For isolation of brown fat-derived SVF, the iBAT was dissected from 4-week-old WT and *Sirt7* KO mice, minced in the presence of 10 mM CaCl₂, and then digested with 2 mg/mL collagenase type II (Worthington Biochemical) and 2.4 U/mL Dispase II (Roche) in PBS buffer at 37 °C for 50 min with shaking. The cell suspension was filtered through a 70-μm cell strainer to a new tube, and the flow-through cells were centrifuged at $700 \times g$ for 5 min. The cell pellet was resuspended in complete medium (Dulbecco's Modified Eagle's Medium (DMEM)/Nutrient F-12 [1:1] (Gibco BRL) containing 10% fetal bovine serum (FBS) (Chile origin, Biosera), 100 U/mL penicillin (Meiji Seika Pharma), and 100 μg/mL streptomycin (Meiji Seika Pharma)) and then seeded onto a cell culture plate (day 0). For differentiation into brown adipocytes, cells grown to 100% confluence (day 1) were exposed to induction medium (complete medium containing 625 μM isobutylmethylxanthine (Sigma-Aldrich), 125 μM indomethacin (Sigma-Aldrich), 5 μM dexamethasone (Sigma-Aldrich), 8.5 nM insulin (Fujifilm Wako Pure Chemical Corporation), and 1 nM T3 (Sigma-Aldrich)). Two days after induction, the cells were maintained in differentiation medium (complete medium containing 8.5 nM insulin and 1 nM T3), and the medium was changed every 2 days until the cells were ready for harvest or an OCR assay (day 6–7 post-differentiation).

HEK293T cells (#632180, Clontech) were cultured in DMEM (Fujifilm Wako Pure Chemical Corporation) with 10% FBS (Dominican Republic origin, Biosera). 293AAV cells (#AAV-100, Cell Biolabs) were cultured in DMEM with 10% FBS (Dominican Republic origin, Biosera), 0.1 mM MEM Non-Essential Amino Acids (Fujifilm Wako Pure Chemical Corporation), 2 mM GlutaMAX-1 supplement (Gibco), 100 U/mL penicillin, and 100 μg/mL streptomycin.

### Metabolic cage studies

Mice were housed individually in metabolic cages with unlimited access to water and food and were maintained on a 12-h light/dark cycle at 24 °C or 30 °C. The mice were acclimated in the metabolic cages for 2 days before the start of the experiments. The VO₂, VCO₂, locomotor activity, food/water intake, and body weight were monitored with an O₂/CO₂ metabolic measuring system (Model MK-5000, Muromachi Kikai, Japan) for 2 consecutive days. The RER is the ratio of carbon dioxide production to oxygen consumption (VCO₂/VO₂). The abbreviated Weir equation was used to calculate the 24-h energy expenditure (1.44 [3.9 × VO₂ (mL/min) + 1.1 × VCO₂ (mL/min)])[65].

### Histological studies

Excised iBAT samples were fixed in 10% (v/v) neutrally buffered formalin (Fujifilm Wako Pure Chemical Corporation) overnight at 4 °C, embedded in paraffin, and cut into 4-μm slices for hematoxylin and eosin (H&E) staining. Lipid areas were quantified by using ImageJ (NIH).

### Body temperature

Mice were hand-restrained and a rectal probe (RET-3, Physitemp) was gently inserted into the rectum to a depth of 2 cm. The rectal temperature was monitored by using an electronic thermistor (Model BAT-12, Physitemp). To alleviate acute stress-induced increases in body temperature, the mice were trained in advance to the measurement procedure and to the restraint every day for 1 week.

### Daily torpor experiment

Daily torpor experiments were performed according to a previously established protocol[42]. Mice were individually housed in metabolic cages placed in a chamber at 24 °C with unlimited access to water and food and were maintained for 2 days on a 12-h light/dark cycle to acclimate to the metabolic cages. Then, the temperature of the chamber was changed to 20 °C, and the VO₂ and VCO₂ were monitored for 3 days. To induce daily torpor, food was removed on the second

day. A typical daily torpor pattern started after 20 h of fasting. Body temperature was monitored after 24 h fasting.

## Gene expression studies

Total RNA was extracted by using Sepasol RNA I super reagent (Nacalai Tesque), and cDNA synthesis was then achieved with a Prime Script RT Reagent Kit (TaKaRa). Real-time qPCR was performed with SYBR Premix Ex Taq II (TaKaRa) and an Applied Biosystems 7300 or ViiA7 thermal cycler (Thermo Fisher Scientific). The relative expression of each gene was normalized to that of *ribosomal protein L19* (*Rpl19*). The primer sequences are listed in Supplementary Table 1.

## Western blotting

Total lysates of cells and organs were obtained by lysis in RIPA buffer (50 mM Tris-HCl (pH 8.0), 150 mM NaCl, 0.1% SDS, 1% NP-40, 5 mM EDTA, and 0.5% sodium deoxycholate) with a protease inhibitor cocktail (PIC) (Nacalai Tesque). For western blotting, proteins were separated by SDS- PAGE and transferred to a PVDF membrane (Immobilon-P, Millipore), which was probed with the primary antibodies. After incubation with the secondary antibodies, proteins were visualized by using Chemi-Lumi One Super (Nacalai Tesque) and a ChemiDoc Imaging System (Bio-Rad Laboratories). The protein bands were quantified by Image Lab software (Bio-Rad). The primary/secondary antibodies are listed in Supplementary Table 2.

## Evaluation of mtDNA copy number

mtDNA copy number was evaluated by determining the ratio of mtDNA to nuclear DNA (nDNA). iBAT or differentiated brown adipocytes were digested with 300 μg/mL proteinase K (Nacalai Tesque) in DNA lysis buffer (100 mM Tris-HCl pH 8.5, 5 mM EDTA, 0.2% SDS, 200 mM NaCl) at 55 °C overnight with shaking, followed by extraction using protein precipitation solution (Promega) and phenol/chloroform. mtDNA was amplified using primers specific for the NADH dehydrogenase 1, mitochondrial (*mt-Nd1*) gene and normalized to nDNA by amplification of the platelet/endothelial cell adhesion molecule 1 (*Pecam1*). The primer sequences are listed in Supplementary Table 1.

## Quantification of T3 by enzyme-linked immunosorbent assay (ELISA)

iBAT was collected and weighed following blood removal by perfusion with PBS and was immediately frozen in liquid nitrogen. Then, the iBAT was homogenized in buffer containing 0.1 M phosphate (pH 7.0), 1 mM EDTA, and 4 mM dithiothreitol by sonication (Sonifier-150, Branson) at 4 °C. After centrifugation at 5000 × *g* for 5 min at 4 °C, homogenates were used for the measurement of tissue T3 content. Levels of serum and tissue T3 were measured by using a T3 (Total) (Mouse/Rat) ELISA Kit (KA0925, Abnova) and microplate reader (iMark 168-1130, BIO-RAD) according to the manufacturer's instructions.

## Oil red O staining

Differentiated brown adipocytes were washed three times with PBS, fixed with 10% (v/v) neutral buffered formalin for 10 min, washed two times with PBS, treated with 60% isopropanol, and incubated with filtered oil red O (0.18% Oil red O in 60% isopropanol; Sigma-Aldrich) for 20 min. After washing with 60% isopropanol and subsequent PBS washes, lipid droplet images were captured with a BZX-700 microscopy system (Keyence). Stained oil red O was re-extracted in 100% isopropanol for 5 min, and lipid accumulation was semi-quantitatively measured by the OD492. Background control comprised 100% isopropanol.

## Extracellular flux measurement

The OCR was measured using the XF24 Extracellular Flux Analyzer (Seahorse Bioscience). SVF cells (5 × 10^5 cells/well) were plated in an XF24-well cell culture microplate (Seahorse Bioscience) and differentiated into brown adipocytes, followed by OCR measurement at 37 °C according to the manufacturer's instructions. Briefly, differentiated cells were incubated in pre-warmed assay medium (Seahorse XF base medium (Agilent Technologies) with 25 mM glucose, 1 mM sodium pyruvate, 2 mM L-glutamine) for 30 min without $CO_2$, and a mitochondrial stress test was conducted. To detect the uncoupled respiration, maximal respiration, and non-mitochondrial respiration, 5 μM oligomycin, 10 μM FCCP, and 0.5 μM rotenone/antimycin (Agilent Technologies) were injected, respectively. The OCR was normalized to the lipid accumulation (OD492 value), and the data are expressed as pmol of $O_2$/min/OD492.

## Plasmid construction

pFN18A-SIRT7, pcDNA3-FLAG-SIRT7, and pcDNA3-FLAG-SIRT7^H188Y were generated previously[35]. pFN18A (Promega), pAAV-DJ/8 (Cell Biolabs), pHelper (Cell Biolabs), pCMVp300-myc (#30489, Addgene), pX602-AAV-TBG::NLS-SaCas9-NLS-HA-OLLAS-bGHpA;U6::Bsal-sgRNA (#61593, Addgene), pBIND (Promega), pGL3-Basic (Promega), and pTD1 (Shimadzu) were purchased. To generate pcDNA3.1-3×HA-IMP2, mouse *Imp2* cDNA was amplified by PCR (the primers used are listed in Supplementary Table 1), verified by sequencing, and then cloned into pcDNA3.1-3×HA (double-stranded oligo DNA of the HA tag was ligated into pcDNA3). For GAL4DBD fusion IMP2 expression, fragments of IMP2 (amino acid residues 1–200 [A], 201–380 [B], 381–592 [C], 381–494 [C1], 495–592 [C2], 381–481 [C1A], and 442–494 [C1B]) were amplified by PCR, cloned into the pBIND, and verified by sequencing. Various KR and KQ mutants of IMP2 were introduced using a KOD-plus Mutagenesis Kit (TOYOBO) and verified by sequencing. To generate pFN18A-IMP2, pFN18A-IMP2^K438R, and pFN18A-IMP2^K438Q, the EcoRI/XbaI-digested fragments of pcDNA3.1-3×HA-IMP2, pcDNA3.1-3×HA-IMP2^K438R, and pcDNA3.1-3×HA-IMP2^K438Q were subcloned into the pFN18A, respectively. Long *Ucp1* 3′-UTR (Ucp1L: 485 bp), short *Ucp1* 3′-UTR (Ucp1S: 78 bp), and a partial *Ucp1* CDS (Ucp1C: a 524-bp sequence spanning 336–859 bp downstream of the transcription start site) were cloned into the pTD1 vector (Shimadzu) to generate pTD1-Ucp1L, pTD1-Ucp1S, and pTD1-Ucp1C, respectively.

## HaloTag pulldown assay

Either pFN18A (for Halo) or pFN18A-SIRT7 (for Halo-SIRT7) was transformed into *E. coli* K12 (Single Step (KRX) Competent Cells, Promega), and each *E. coli* culture was cultured at 37 °C until the OD600 reached 0.5. After further culture at 20 °C until the OD600 reached 0.7, 20% rhamnose (Nacalai Tesque) was added at a 1:200 dilution and the cells were cultured overnight. Following centrifugation at 1400 × *g*, the pellet was resuspended in Halo purification buffer (50 mM HEPES-KOH (pH 7.4), 150 mM NaCl, 1% NP-40, 1 mM PMSF, and PIC). Then, the cells were disrupted by three freeze–thaw cycles, followed by two sonications for 20 s each at level 2 (Sonifier-150). The cell lysate was centrifuged at 6000 × *g* for 10 min, and the resulting supernatant was filtered through a 0.45-μm filter. Cleared lysate was incubated with HaloLink Resin (Promega) for 2 h at room temperature. After binding, the resin was washed five times with the same Halo purification buffer to obtain purified HaloLink Resin-bound Halo or Halo-SIRT7.

Halo or Halo-SIRT7 proteins immobilized on HaloLink Resin were incubated with cell lysate in pull-down buffer (10 mM Tris-HCl (pH 7.4), 150 mM NaCl, 0.5% NP-40, 10 mM NaF, 10 mM $Na_4P_2O_7$, 1 mM PMSF, and PIC) containing 10 mM nicotinamide (Sigma-Aldrich) and 1 μM TSA (Fujifilm Wako Pure Chemical Corporation). After incubation at 4 °C overnight for binding, the resins were washed five times with the pull-down buffer. The bound proteins were detected by western blotting with the respective antibody. For comprehensive identification of protein, the bound proteins were resolved by SDS-PAGE, followed by silver staining and mass spectrometry analysis.

## Mass spectrometry (MS) analysis

Extracted proteins were separated by SDS-PAGE and stained by using the Proteo Silver Plus Silver Stain Kit (Sigma-Aldrich). A sample lane of the gel was cut into 26 small pieces, and destained according to the manufacturer's protocol. Gel pieces were washed, dried in a Savant SpeedVac System (Thermo Fisher Scientific), and reduced with 10 mM dithiothreitol at 56 °C for 1 h. After the gel pieces were alkylated with 55 mM iodoacetamide under agitation in the dark at room temperature for 45 min, the supernatant was removed. The gel pieces were washed and re-dried. After the gel pieces were incubated with 50 µg/mL trypsin (V5113, Promega) overnight at 37 °C, the eluted mixture was dried and dissolved in 40 µL of MS-grade water containing 2% acetonitrile (Wako Pure Chemical Industries) and 0.1% trifluoroacetic acid (Kanto Chemical). The peptide solution was analyzed by a LC-MS/MS system (LTQ Velos Pro, Thermo Fisher Scientific) as previously reported[35]. The peptide mixture was injected into an L-trap column (0.3 × 5 mm, 5 mm; Chemical Evaluation Research Institute) and separated by a capillary reversed-phase C18 column (0.1 × 150 mm, 3 µm; Chemical Evaluation Research Institute) with gradient elution (solvent A: MS-grade water containing 0.1% formic acid (Kanto Chemical); solvent B: 100% acetonitrile). All MS/MS spectra were searched against *Mus musculus* entries in the Swiss-Prot database (v2013-01-04) using SEQUEST (Proteome Discoverer software, Thermo Fisher Scientific) database search program using Proteome Discoverer 1.3. A false discovery rate <0.05 was considered when searching the results. Precursor and fragment mass tolerances were 2 Da and 0.8 Da, respectively. A maximum of two missed cleavages with trypsin was allowed. Dynamic modification of oxidation of methionine and static modification of carbamidomethyl on cysteine were considered. MS raw data have been deposited at ProteomeXchange Consortium repository jPOSTrepo[66] (https://repository.jpostdb.org/) and are publicly available under accession code JPST001343/PXD028870.

## Co-immunoprecipitation assays

HEK293T cells transfected with the indicated expression plasmids by the jetPRIME reagent (Polyplus) for 24 h were lysed in lysis buffer (20 mM Tris-HCl (pH 7.4), 150 mM NaCl, 2.5 mM MgCl$_2$, 0.5% NP-40, 1 mM PMSF, and PIC) containing 10 mM nicotinamide and 1 µM TSA by six passes through a 29-G needle. After centrifugation at 14,000 × g for 10 min at 4 °C, the cell lysate (500 µg) was subjected to immunoprecipitation overnight at 4 °C with anti-DYKDDDDK (FLAG) tag antibody beads (clone 1E6, Wako Pure Chemical Industries), and the resins were washed five times with lysis buffer. Precipitates were examined by western blotting with the indicated antibodies.

## AAV preparation and infection

Gene disruption in SVF cells was performed using an AAV-based CRISPR/SaCas9 system. An AAV-DJ/8 helper-free system was purchased from Cell Biolabs, Inc. The pAAV-SaCas9-sgRNA vector pX602-AAV-TBG::NLS-SaCas9-NLS-HA-OLLAS-bGHpA;U6::BsaI-sgRNA was a gift from Feng Zhang (#61593, Addgene)[67]. sgRNAs targeting exonic regions of the mouse *Imp2* gene were based on a previous report[68], and a pair of annealed guide oligonucleotides (listed in Supplementary Table 1) was cloned into the BsaI site of the pAAV-SaCas9-sgRNA vector. AAV was produced according to the instructions of the AAV helper-free system. Near-confluent 293AAV cells (Cell Biolabs) in a 10-cm dish were co-transfected with either 2 µg of pAAV-SaCas9-sgRNA vector or pAAV-SaCas9-*Imp2* sgRNA vector, 2 µg of pAAV-DJ/8, and 3 µg of pHelper plasmids using jetPRIME transfection reagent. Three days after transfection, AAV viral particles were purified using AAVpro Purification kit (All Serotypes) (TaKaRa) and virus titration was carried out using the QuickTiter AAV Quantitation Kit (Cell Biolabs) following the manufacturer's instructions. For knockout of *Imp2* in SVF cells, cells (1 × 10⁴ cells/well) were infected with 1 × 10⁵ genome copies/cell of either AAV(DJ/8)-SaCas9-control sgRNA or AAV(DJ/8)-SaCas9-*Imp2*

sgRNA for 24 h. The infected cells were exposed to the induction medium to induce adipogenic differentiation for 2 days and then cultured in differentiation medium for an additional 6–7 days until OCR experiments.

## Detection of lysine acetylation

HEK293T cells transfected with the indicated plasmids by the jetPRIME reagent for 24 h were lysed in IP buffer (20 mM Tris-HCl (pH 7.4), 200 mM NaCl, 2.5 mM MgCl$_2$, 0.2% NP-40, 1 mM PMSF, and PIC) containing 10 mM nicotinamide and 1 µM TSA by sonication (Sonifier-150) at 4 °C. After centrifugation at 14,000 × g for 10 min at 4 °C, the cell lysates (200 µg) and anti-HA tag antibody beads (clone 3F10, Wako Pure Chemical Industries) were incubated overnight at 4 °C. After five washes with IP buffer, the precipitated proteins were eluted with 2 × SDS sample buffer (100 mM Tris-HCl (pH 6.8), 4% SDS, 20% glycerol, and 0.2% bromophenol blue), and lysine acetylation was detected by western blotting with an anti-acetylated-lysine antibody. To detect the endogenous acetylation of IMP2, 1000 µg lysate from iBAT of WT and *Sirt7* KO mice was incubated with anti-IMP2 antibody-crosslinked resin at 4 °C overnight for immunoprecipitation. An anti-IMP2 antibody-crosslinked resin was prepared using a Pierce Crosslink Immunoprecipitation Kit (Thermo Scientific) according to the manufacturer's protocol.

## RNA immunoprecipitation

pTD1-Ucp1L, pTD1-Ucp1S, and pTD1-Ucp1C were linearized by NotI digestion and subjected to RNA synthesis using ScriptMAX Thermo T7 Transcription Kit (TOYOBO). The yielded RNAs were collected using a RNeasy Mini Kit (QIAGEN) and used as RNA input for an immunoprecipitation assay, as described below.

HEK293T cells transfected with pcDNA3.1-3×HA, pcDNA3.1-3×HA-IMP2, pcDNA3.1-3×HA-IMP2$^{K438R}$, or pcDNA3.1-3×HA-IMP2$^{K438Q}$ by the JetPRIME reagent for 24 h were lysed in IP buffer (20 mM Tris-HCl (pH 7.4), 150 mM NaCl, 2.5 mM MgCl$_2$, 0.05% NP-40, 1 mM PMSF, and PIC) containing 10 mM nicotinamide and 1 µM TSA by ten passes through a 27-G needle. After centrifugation, the cleared lysates were subjected to immunoprecipitation overnight at 4 °C with anti-HA antibody beads (clone 4B2, Wako Pure Chemical Industries). The beads were incubated with IP buffer containing 5 ng/µL RNase A (Nippon Gene) at room temperature for 15 min, washed five times with IP buffer, and then incubated in IP buffer containing synthesized RNAs (for Fig. 8e, 2500 ng Ucp1L RNA; for Supplementary Fig. 6d, 1250 ng Ucp1L RNA + 1250 ng Ucp1C RNA or 1250 ng Ucp1S RNA + 1250 ng Ucp1C RNA). IMP2-bound *Ucp1* RNAs were isolated according to the RiboCluster Profiler RIP Assay Kit protocol (MBL) and subjected to real-time qPCR.

## Expression and purification of recombinant IMP2 protein

The recombinant proteins for mouse IMP2 were generated by the HaloTag Protein Purification System. pFN18A-IMP2, pFN18A-IMP2$^{K438R}$, or pFN18A-IMP2$^{K438Q}$ transformed *E. coli* K12 cells were cultured as described above. The cell lysate was treated with 5 mM ATP (Sigma-Aldrich) and 20 mM MgCl$_2$ before being incubated at 37 °C for 10 min and then centrifuged at 6000 × g for 10 min. The resulting supernatant was filtered through a 0.45-µm filter, and the cleared lysate was incubated with HaloLink Resin for 2 h at room temperature. After binding, the resin was washed three times with Halo purification buffer and two times with Halo purification buffer without NP-40 and PMSF. ProTev Plus protease (Promega) was used to separate each IMP2 from the HaloTag, and the buffer of the eluted protein solution was changed to PBS.

## In vitro translation

In vitro translation was performed using an insect cell-free protein synthesis system, Transdirect insect cell (Shimadzu). The coding

sequence of firefly *luciferase* RNA (from pGL3-Basic (Promega)) was fused to the murine *Ucp1* 5′-UTR and long 3′-UTR (amplified by PCR and verified by sequencing), and this *luciferase-Ucp1* UTR was introduced into pTD1 vector at a site under the control of the T7 promoter. The mRNA encoding the cDNA was prepared using a ScriptMAX Thermo T7 Transcription Kit according to the manufacturer's instructions. The mixture (25 μL of insect cell extract, 15 μL of reaction buffer, 1 μL of 4 mM methionine, 4 μL of mRNA (4 ng)) was divided into five equal aliquots (9 μL each) and incubated with 1 μL recombinant IMP2 proteins (20 ng) at 25 °C for 5 h. The translated luciferase polypeptide was estimated by the gain in luciferase activity using a dual-luciferase reporter assay system (Promega) according to the manufacturer's instructions.

### Statistics and reproducibility

No statistical methods were used to determine sample size, but the sample sizes were similar to those of previous reports[33,35]. No exclusion/inclusion criteria were applied to the mice used in this study. All data were analyzed using appropriate statistical methods with GraphPad Prism 9 (GraphPad software). All results are expressed as the mean ± SEM. Statistical significance was tested using the two-tailed Student's *t*-test for comparison between two groups. Two-way ANOVA with Bonferroni's multiple comparisons test was used for body temperature, VO$_2$, and energy expenditure. In all analyses, $p < 0.05$ was considered to indicate a significant difference. All experiments were replicated by at least two independent experiments unless otherwise noted, and details on sample numbers and statistics of each experiments were described in figures or figure legends.

### Reporting summary

Further information on research design is available in the Nature Portfolio Reporting Summary linked to this article.

## Data availability

The mass spectrometry proteomics data used in Fig. 7a have been deposited in the ProteomeXchange Consortium repository jPOSTrepo[66] (https://repository.jpostdb.org/) and are publicly available under accession code JPST001343/PXD028870. The reporting summary for this article is available in the Supplementary Information section. All the other data supporting this study are available within this Article, Supplementary Information, and Source Data. Source data are provided with this paper.

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

## Acknowledgements

We thank the members of Yamagata Laboratory for discussions and technical assistance. We thank Dr. Evan D Rosen (Beth Israel Deaconess Medical Center and Harvard Medical School), Dr. Wataru Ogawa (Kobe University), and Dr. Jun Eguchi (Okayama University) for providing *Ucp1-Cre* and *Adipoq-Cre* transgenic mice. This study was supported by Grants-in-Aid for Scientific Research (B) (19H03711: K.Y.; 20H04107: T.Y.) and Challenging Research (Exploratory) (19K22639: K.Y.); by a grant from the Japan Agency for Medical Research and Development under Grant Number (JP21gm5010002: K.Y.); and by grants from the Naito Foundation (K.Y.) and Takeda Science Foundation (K.Y., T.Y.). The research in the J.A. laboratory was supported by the EPFL, the EU Ideas program (ERC-AdG-231138), and the SNSF (31003A_179435).

## Author contributions

T.Y. and K.Y. conceived the study. T.Y. designed the experiments and wrote the manuscript. T.Y. performed most of the animal experiments. Y.S. performed statistical analyses and the primary brown adipocyte experiments. S.U.S. and T.Y. performed the biochemical experiments. M.T. and N.A. conducted mass spectrometry analysis. T.M., T.T., Md.F.K., K.M., M.Y., and Y.K. carried out some of the experiments, analyzed some data, and provided useful suggestions. E.A., S.K., Y.O., T.B., E.V., and J.A. provided critical supplies and support for experiments.

## Competing interests

The authors declare no competing interests.
