## [Peer Review File · Nature Communications]

SIRT7 suppresses brown adipose tissue thermogenesis in miceREVIEWER COMMENTS

Reviewer #1 (Remarks to the Author):

The manuscript entitled "SIRT7 serve as an energy-saving factor by suppressing brown fat thermogenesis" by Tatsuya Yoshizawa et al. investigates the role of Sirt7 as a thermogenesis brake. This function contrasts with the positive roles of other Sirt family members in brown adipose tissue and browning. The main novelty is the identification of SIRT7 as an indirect regulator of UCP1 translation through the deacetylation of IGF2BP2.

The conclusions of this work are interesting and potentially relevant in the field, especially considering the previously identified roles of sirtuins in brown adipose tissue. However, we feel that the manuscript falls short in the characterisation of some of the models.

Major comments:

- The authors use three different models to characterise the effects of Sirt7: a whole-body knockout, an adipocyte-specific knockout and a brown adipocyte-specific knockout. Although they all exhibit the same temperature and energy expenditure defects, there are several differences in the expression of critical genes in BAT (Figure 4a, d, e). Differences in UCP1 protein levels are also more prominent in the whole-body KO than in the other models. These differences may indicate additional effects of Sirt7 in other organs that affect the activation levels of the brown adipose tissue in these animals indirectly. Further experiments at thermoneutrality, a condition where the sympathetic stimulation and other external cues are minimal, would help to strengthen the conclusions.

- Dio2 and Clstn3b are not upregulated in the brown adipocyte-specific mouse model (Figure 4e), but they are upregulated in the adipocyte and whole-body KO. Have the authors measured iBAT T3 levels in this model? Do they have any explanation for these differences?

- The panel of genes analysed should be expanded (e.g. Ebf2, Cebpb, Cidea, Bmp8b...). The whole set of genes tested in Figure 4a should also be included in the adipocyte- and brown adipocyte-specific models.

- The contribution of Sirt7 to browning in these models is not clear. The results in Supplementary Figure 4b are not statistically significant, and they are only referring to the whole-body knockout. Histological analyses and cold exposure experiments may help to clarify the phenotype. It is also not indicated if browning has been evaluated in the adipocyte-specific model. Independently of this result, the reasons for the differential effect of Sirt7 absence in brown and beige adipose tissue should be discussed in the manuscript.

- The interpretation of the Seahorse experiment in Figure 5e is not accurate. Differences in the

uncoupling capacity are usually observed in the basal respiration or the proton leak (this is in fact observed in Figure 6d). Differences in the maximal respiratory capacity suggest alterations in the number or the efficiency of the mitochondria.

- All the experiments described in the manuscript have been performed in males. If the mouse model is still available, at least a basic characterisation of the phenotype should also be performed in females.

Minor comments:

- Supplementary figure 3b is described as showing statistical differences in the weight of epiWAT and iBAT, but there are no differences in epiWAT according to the graph. Please, correct this error in the text.

- Immunoblot panels in Figures 7a, b and d would benefit from including quantifications and statistical analyses.

Reviewer #2 (Remarks to the Author):

Yoshizawa et al.

“SIRT7 serves as an energy-saving factor by suppressing brown fat thermogenesis”

This study by Yoshizawa et al. reports that SIRT7 has a role in the regulation of energy metabolism and thermogenesis through the regulation of UCP-1 protein levels in brown adipose tissue (BAT) in mice. They show that whole-body Sirt7 KO mice, as well as adipose tissue-specific and BAT-specific Sirt1 KO mice, maintain higher body temperature and energy expenditure compared to control mice under an ambient temperature. Whole-body Sirt7 KO mice also maintain their energy expenditure and body temperature during daily torpor and aging. The authors further dissected the molecular mechanism by which SIRT7 regulates UCP-1 protein levels and found that SIRT7 deacetylates insulin-like growth factor 2 mRNA binding protein 2 (IGF2BP2/IMP2), an RNA-binding protein that inhibits the translation of Ucp1 mRNA, at K438.

Although the results presented in this study look convincing and important, the biggest problem in this study is whether the phenotypes of body temperature and energy expenditure could be explained by the same mechanism in BAT. As described in detail below, the observed changes in UCP-1 protein levels in SIRT7 KO BAT are an unlikely explanation for increased energy expenditure. To rigorously address this critical issue, the authors should test their animals at thermoneutrality (~30C for mice). The IMP2-UCP-1 axis is most likely a reasonable explanation for increased body temperature under an ambient condition, and the authors might want to limit their presentation to this particular phenotype. But if that is the case, the novelty of the study would be dampened. Therefore, the authors are strongly encouraged to

look deeper into the systemic interaction between adipose tissue and other tissues, such as the liver and skeletal muscle, to fully explain the mechanism for increased energy expenditure.

Major comments:

1) It has been known that the contribution of adipose tissue to whole-body energy expenditure is relatively very small and also that UCP-1 knockout mice show no differences in energy expenditure under an ambient condition. It is likely that increased BAT function through increased UCP-1 levels could provide a reasonable explanation for increased body temperature. Nonetheless, increased body temperature would not necessarily lead to increased energy expenditure. Given significant increases in energy expenditure throughout light and dark times in whole-body, Ad, and BAd SIRT7 KO mice, it is very likely that other major organs, such as the liver and skeletal muscle, are involved in these observed changes in energy expenditure.

It would be ideal if the authors could conduct further analyses to obtain a more reasonable and convincing explanation for the energy expenditure phenotype. However, this may not be easy to do with a limited amount of time and resource for revision. Therefore, a minimal requirement would be to test whole-body and BAd SIRT7 KO mice at thermoneutrality. This particular experiment could provide more convincing results for the importance of SIRT7-dependent UCP-1 protein regulation in BAT.

2) For the same reason described above, it remains unclear whether “the importance of SIRT7 in the suppression of energy expenditure and thermogenesis is much higher in aged mice.” In aged mice, is there any change in SIRT7 protein amounts or SIRT7 activity?

Minor comments:

1) A quantification and scale bar should be provided in Fig. 1k.

2) The authors mentioned that VO₂, energy expenditure, and body temperature clearly declined in WT mice during aging in page 7, line 162-165. However, there is no statistical analysis for that. Those analysis should be performed in Fig. 2a-c.

3) Although a previous paper shows that Imp2 KO mice increase UCP-1 protein levels in BAT, it is worth checking UCP-1 mRNA and protein level in Sirt7 and Imp2 KO primary brown adipocytes in Fig. 6.

4) In Fig. 1l, are there any clues why there is no difference in body temperature some time points, particularly during the dark time? How about UCP1 protein levels at those times? Although the authors do not show the oscillation of body temperature of adipose tissue- or brown adipocytes-specific Sirt7 KO mice (Fig. 3), it could be meaningful to compare the body temperature oscillation in these three KO models. Interestingly, there are less or no differences in VO₂ around ZT12 between WT and KO in each KO model mice (Fig. 1c, 3l, 3f), similar to the oscillation of body temperature shown in Fig. 1l. Sirt7 might have a role at specific time points, but not throughout a day. If UCP1 protein levels in BAT still

showed a difference between WT and Sirt7 KO mice around ZT12, using an infrared camera would be a good idea to examine where the heat comes from.

5) Tables or figures should be provided for the mass spectrometry experiment including values and other candidates in Fig. 6.

6) Page 12, line 275-277, the authors mentioned a Western blotting experiment, however there is no result in Fig. 6.

Reviewer #3 (Remarks to the Author):

The manuscript by Yoshizawa et al. describes a new mechanism of Sirt7 by deacetylating IMP2 at K438 to inhibit Ucp1 translation. This study started with thorough characterization of Sirt7-KO mice (2 independent lines in young and old ages) in body weight, energy expenditure, body temperature, food intake... etc. The authors then used adipocyte-specific (AdKO) and BAT-specific (BAdKO) Sirt7-KO to validate the essential role of Sirt7 in suppressing Ucp1 protein level and BAT thermogenesis. They further identified that the RNA-binding protein IMP2 as a substrate of Sirt7, whose deacetylation at K438 residue is important to inhibit Ucp1 translation. Overall, the main conclusions in this manuscript are well supported by the presented results from a combination of in vitro (cell culture and reporter assay) and in vivo (various global and conditional KO of Sirt7) experiments. The most part of this work is very solid and no major conceptual or experimental flaws, but there are some aspects with room for improvement as described below.

Specific Comments:

1) In the introduction- line 90, about posttranscriptional regulation of Ucp1 mRNA, 2 more papers, Takahashi et al 2015, Cell Rep. 13:2756 and Chen et al 2018 EMBO J 37:e99071, should be included.

2) Lines 102-105, "SIRT7 is a relatively unique sirtuin. Its enzymatic activities and functional roles had been only partially elucidated until recently, but recent studies have revealed many biological functions for SIRT7," What does it mean about the relative uniqueness of Sirt7 in comparison to other sirtuins? "had been only partially elucidated until recently, but recent studies....." is an awkward expression, please rephrase it.

3) In lines 117-118, "We also demonstrate acetylation-dependent of Ucp1 mRNA by RNA-binding proteins (RBP)." Because only IMP2 was studied here, "RNA-binding proteins" refer to? Any other RNA-binding proteins, such as CPEB2, BRF1 or those listed in the Supp Fig 5d, were identified from Halo-tag-Sirt7 pull-down experiments (Fig. 6a)?

4) The authors appeared to generate floxed Sirt7 allele in C57BL/6J mice but they crossed the floxed mice with adipoq-cre (JAX010803) and Ucp1-cre (JAX024670) mice in FVB/N background. Because genetic background can influence various metabolic parameters, how does the authors control the variations caused by mixed genetic background?

5) In Fig 4b, 4f, 4g, the immunoblotting of Sirt7 should be included. It is important to show the depletion of Sirt7 in iBAT.

6) The previous study (EMBO J 37:e99071) reported that alternative polyadenylation produces Ucp1 transcripts carrying long (~10%) or short (~90%) 3'-UTR in mouse BAT. Does deacetylation/acetylation of IMP2 affect translation of both long and short forms of Ucp1 transcripts? In Fig 7e, which kind of 3'-UTR was used for the reporter assay, the majority short form or the minority long form? The authors should test IMP2 and its deacetylation/acetylation-mimetic mutants in the luciferase reporters appended with long and short Ucp1 3'-UTR. Moreover, they should also demonstrate that K438R but not K438Q mutant binds to Ucp1 mRNA. Please also specify which Ucp1 3'-UTR (long, short or both) was bound by IMP2.

Report of Reviewer #1

The manuscript entitled “SIRT7 serve as an energy-saving factor by suppressing brown fat thermogenesis” by Tatsuya Yoshizawa et al. investigates the role of Sirt7 as a thermogenesis brake. This function contrasts with the positive roles of other Sirt family members in brown adipose tissue and browning. The main novelty is the identification of SIRT7 as an indirect regulator of UCP1 translation through the deacetylation of IGF2BP2.

The conclusions of this work are interesting and potentially relevant in the field, especially considering the previously identified roles of sirtuins in brown adipose tissue. However, we feel that the manuscript falls short in the characterisation of some of the models.

Major comments:

- The authors use three different models to characterise the effects of Sirt7: a whole-body knockout, an adipocyte-specific knockout and a brown adipocyte-specific knockout. Although they all exhibit the same temperature and energy expenditure defects, there are several differences in the expression of critical genes in BAT (Figure 4a, d, e). Differences in UCP1 protein levels are also more prominent in the whole-body KO than in the other models. These differences may indicate additional effects of Sirt7 in other organs that affect the activation levels of the brown adipose tissue in these animals indirectly. Further experiments at thermoneutrality, a condition where the sympathetic stimulation and other external cues are minimal, would help to strengthen the conclusions.
- Dio2 and Clstn3b are not upregulated in the brown adipocyte-specific mouse model (Figure 4e), but they are upregulated in the adipocyte and whole-body KO. Have the authors measured iBAT T3 levels in this model? Do they have any explanation for these differences?
- The panel of genes analysed should be expanded (e.g. Ebf2, Cebpb, Cidea, Bmp8b...). The whole set of genes tested in Figure 4a should also be included in the adipocyte- and brown adipocyte-specific models.
- The contribution of Sirt7 to browning in these models is not clear. The results in

Supplementary Figure 4b are not statistically significant, and they are only referring to the whole-body knockout. Histological analyses and cold exposure experiments may help to clarify the phenotype. It is also not indicated if browning has been evaluated in the adipocyte-specific model. Independently of this result, the reasons for the differential effect of Sirt7 absence in brown and beige adipose tissue should be discussed in the manuscript.

- The interpretation of the Seahorse experiment in Figure 5e is not accurate. Differences in the uncoupling capacity are usually observed in the basal respiration or the proton leak (this is in fact observed in Figure 6d). Differences in the maximal respiratory capacity suggest alterations in the number or the efficiency of the mitochondria.

- All the experiments described in the manuscript have been performed in males. If the mouse model is still available, at least a basic characterisation of the phenotype should also be performed in females.

Minor comments:

- Supplementary figure 3b is described as showing statistical differences in the weight of epiWAT and iBAT, but there are no differences in epiWAT according to the graph. Please, correct this error in the text.

- Immunoblot panels in Figures 7a, b and d would benefit from including quantifications and statistical analyses.

Responses to the comments of Reviewer #1

We wish to thank the reviewer for the comment, “The conclusions of this work are interesting and potentially relevant in the field, especially considering the previously identified roles of sirtuins in brown adipose tissue.”, and for his/her constructive suggestions.

In response to your comment, “the manuscript falls short in the characterisation of some of the models. ”, we have more deeply characterized all three mouse models. We have addressed all of the points raised by the three reviewers through new experiments and/or new text. Consequently, we have reorganized the text and figures to include the additional findings, particularly those concerning the systemic interaction between BAT and other tissues.

Major comments:

“The authors use three different models to characterise the effects of Sirt7: a whole-body knockout, an adipocyte-specific knockout and a brown adipocyte-specific knockout. Although they all exhibit the same temperature and energy expenditure defects, there are several differences in the expression of critical genes in BAT (Figure 4a, d, e). Differences in UCP1 protein levels are also more prominent in the whole-body KO than in the other models. These differences may indicate additional effects of Sirt7 in other organs that affect the activation levels of the brown adipose tissue in these animals indirectly. Further experiments at thermoneutrality, a condition where the sympathetic stimulation and other external cues are minimal, would help to strengthen the conclusions.”

The reviewer is correct. We also believe that SIRT7 in other organs affects the gene expressions of BAT in whole-body *Sirt7* KO mice, and greatly appreciate the recommendation that further experiments be performed at thermoneutrality (also recommended by reviewer #2). As suggested, we performed the thermoneutral experiments in *Sirt7* KO and *Sirt7* BAdKO mouse lines. As shown in Fig. 3f and g, the phenotype of *Sirt7* KO mice comprising higher VO₂ and energy expenditure was remarkably attenuated at thermoneutrality but was still significantly higher during the active (dark) phase only. The body temperature of *Sirt7* KO mice was still slightly but significantly higher than that of WT mice (Fig. 3h). Therefore, we now conclude that SIRT7 suppresses energy expenditure and thermogenesis via multiple pathways, largely in BAT, but also in other organs. We have added this information to page 10, paragraph 1. In the case of *Sirt7* BAdKO mice at thermoneutrality, the phenotype of higher VO₂, energy expenditure, and body temperature was almost completely abolished (Fig. 5f, g

and Supplementary Fig. 5j). These results strengthen the conclusion that brown adipocytic SIRT7 suppresses whole-body energy expenditure and thermogenesis. We have added this information to page 12, paragraph 1.

Although we had already discussed that white adipocytic SIRT7 indirectly regulates *Dio2* and *Clstn3b* expression in iBAT and mentioned recent studies that found that A-FABP, FGF6, and FGF9, as adipokines, regulate the expression of BAT-related genes such as *Dio2* (in the original manuscript on page 15, paragraph 1), we have now extended the Discussion according to the results of the experiments at thermoneutrality (page 17, paragraph 1) as follows: “Similarly, *Ucp1* expression was significantly increased in *Sirt7* KO mice, but not in the other models. *Ucp1* gene expression is highly induced by the sympathetic stimulation, and the expression levels of *Ucp1* in iBAT of *Sirt7* KO mice at thermoneutrality was no longer high (Supplementary Fig. 3h), suggesting that brain SIRT7 may control *Ucp1* expression in iBAT via the sympathetic nervous system.”

Furthermore, we analyzed SIRT7 protein levels in *Sirt7* AdKO and *Sirt7* BAdKO mice to show the depletion of SIRT7 in iBAT (in response to the suggestion of reviewer #3). As shown in Fig. 4f and 5e, the depletion of SIRT7 was not perfect in *Sirt7* AdKO mice and was more inefficient in *Sirt7* BAdKO mice. Therefore, we have added the following consideration to the Discussion (page 17, paragraph 1): “Moreover, it is probable that a variation in the depletion efficiency of SIRT7 by different Cre drivers (Fig. 4f, 5e) affected these observed changes in gene expression.”

“Dio2 and Clstn3b are not upregulated in the brown adipocyte-specific mouse model (Figure 4e), but they are upregulated in the adipocyte and whole-body KO. Have the authors measured iBAT T3 levels in this model? Do they have any explanation for these differences?”

To address this question, we analyzed iBAT T3 levels in *Sirt7* AdKO and *Sirt7* BAdKO mice. The T3 level was increased in the iBAT of *Sirt7* AdKO mice (Fig. 4e), but not in that of *Sirt7* BAdKO mice (Supplementary Fig. 5i). We have added this information to page 11, paragraphs 1 and 2. As mentioned above, these differences have been discussed on page 17, paragraph 1.

“The panel of genes analysed should be expanded (e.g. *Ebf2*, *Cebpb*, *Cidea*, *Bmp8b*...). The whole set of genes tested in Figure 4a should also be included in the

adipocyte- and brown adipocyte-specific models.”

We thank the reviewer for this comment. To address this comment, we expanded the panel of genes (*Cebpb*, *Ebf2*, *Cidea*, *Cox3*, *Cox8b*, *Bmp8b*, *Fgf21*, *Nrg4*, *Il-6*, *Mstn*) in the *Sirt7* KO mouse line (Fig. 3a, e) (from page 8, paragraph 2 to page 9, paragraph 1; from page 9, paragraph 3 to page 10, paragraph 1). The whole set of genes tested in *Sirt7* KO mice was also included in the *Sirt7* AdKO and *Sirt7* BAdKO mice (Fig. 4d and 5d) (page 11, paragraphs 1 and 2). We found that the expression of the batokines *Fgf21* and *Nrg4* in iBAT was highly elevated in KO types of all three lines. These data suggest that brown adipocytic SIRT7 regulates whole-body energy expenditure through not only BAT thermogenesis, but also endocrine action. We have added this consideration to the Discussion (from page 17, paragraph 2 to page 18, paragraph 1).

“The contribution of *Sirt7* to browning in these models is not clear. The results in Supplementary Figure 4b are not statistically significant, and they are only referring to the whole-body knockout. Histological analyses and cold exposure experiments may help to clarify the phenotype. It is also not indicated if browning has been evaluated in the adipocyte-specific model. Independently of this result, the reasons for the differential effect of *Sirt7* absence in brown and beige adipose tissue should be discussed in the manuscript.”

We thank the reviewer for pointing this out. In the original study, we focused mainly on classical brown adipocytes and did not further examine beige adipocytes. This time, in response to these suggestions, we have examined the contribution of SIRT7 to browning in *Sirt7* AdKO mice. The expression of *Ppargc1a* and *Elovl3* tended to be high in inguinal WAT of *Sirt7* AdKO mice (Supplementary Fig. 4i). Because the browning phenotype is not clear at room temperature (23°C), we administered 1 mg/kg/day norepinephrine to mice for 5 days to induce WAT browning. The results clearly showed that SIRT7 deficiency enhances the browning of inguinal WAT (Supplementary Fig. 4j). We have added this information to page 11, paragraph 1.

Brown and beige adipocytes are different cells that derived from the different precursor cells and have distinct gene expression patterns. Because sirtuin regulates various targets in a cell-dependent manner, its roles vary among organs. For example, SIRT1 gain-of-function induces browning of WAT *in vivo* by deacetylating PPAR γ at K293 and K268, whereas no effects of SIRT1 gain- or loss-of-function are detected in iBAT⁵². Therefore, the differential effects of *Sirt7* absence in brown and beige adipose tissue are reasonable. We have added this consideration to the Discussion (page 18,

paragraph 2).

Moreover, we have conducted cold exposure experiments in response to your suggestion. However, *Sirt7* AdKO mice did not show a clear browning phenotype (*Pparg1a* and *Dio2* had the opposite trend) (Figure 1 for reviewer #1). We have no adequate explanation for why the browning phenotype was not clear under cold exposure. Cold stress regulates several pathways, not only those of the sympathetic nervous system, to activate BAT thermogenesis. Such other signaling pathways in iWAT may be defective in *Sirt7* AdKO mice. Therefore, we finally chose simple norepinephrine treatment.

Figure 1 for reviewer #1.

Real-time qPCR analysis of BAT-related genes in iWAT of 20-week-old male control and *Sirt7* AdKO mice housed for 10 days in a cold environment (12°C).

“The interpretation of the Seahorse experiment in Figure 5e is not accurate. Differences in the uncoupling capacity are usually observed in the basal respiration or the proton leak (this is in fact observed in Figure 6d). Differences in the maximal respiratory capacity suggest alterations in the number or the efficiency of the mitochondria.”

The reviewer is correct. Our discussion of the basal respiration or the proton leak should be changed. To address this concern, we first quantified the basal, maximal, and uncoupled (proton leak) respiratory capacity of the samples in the original Fig. 6d (new Fig. 7d). As shown in Fig. 7e, IMP2 deficiency enhanced the basal, maximal, and uncoupled respiration in WT brown adipocytes, whereas the already increased respiration (basal, maximal, and uncoupled) in *Sirt7* KO cells was not further enhanced by IMP2 disruption. We have added this information to page 13, paragraph 2. Why is the maximal respiratory capacity enhanced in *Imp2*-deficient cells? Previous work demonstrated that IMP2 alters the translation of *Ucp1* mRNA as well as that of 12 other mRNAs encoding mitochondrial proteins, suggesting the importance of IMP2 in mitochondrial functions²¹.

Indeed, *Imp2* KO brown adipocytes showed higher basal, maximal, and uncoupled respiratory capacity in that study²¹. We have added this information to page 13, paragraph 2.

Therefore, we next carefully investigated the experimental conditions to determine why the result of the original Fig. 5e was different from that of the original Fig. 6d. The original Fig. 5e comprised two independent experiments (Figure 2 for reviewer #1). The result of the second experiment, but not that of the first experiment, looks similar to the result of the original Fig. 6d. We confirmed that the differentiation period of brown adipocytes was 5 days in the first experiment and 7 days in the second experiment, while that of the original Fig. 6d experiment was 9 days due to the AAV infection. Because the first experiment was conducted with quite a different protocol, we have decided to remove these data and have performed two more independent experiments with a 7-days differentiation protocol (Figure 3 for reviewer #1). The combined data of the three independent experiments showed that basal and maximal respiratory capacity was significantly higher in *Sirt7* KO cells (Fig. 6e and f) (page 12, paragraph 2). Although the uncoupled respiratory capacity was not significantly different, we adopted this result because repeating the work with a 9-days differentiation protocol would need a lot more mice and would take several more months.

Figure 2 for reviewer #1.

Figure 3 for reviewer #1.

New Figure 6e, f

“All the experiments described in the manuscript have been performed in males. If the mouse model is still available, at least a basic characterisation of the phenotype should also be performed in females.”

We thank the reviewer for pointing out this issue. To address this request, we analyzed female *Sirt7* KO mice. VO_2 and energy expenditure in the dark phase only and body temperature were similarly increased (Supplementary Fig. 1b–d). We have added this information to page 6, paragraph 2.

Minor comments:

“Supplementary figure 3b is described as showing statistical differences in the weight of epiWAT and iBAT, but there are no differences in epiWAT according to the graph. Please, correct this error in the text.”

We apologize for our incorrect explanation in the original manuscript. We have corrected this error in the text (age 10, paragraph 3).

“Immunoblot panels in Figures 7a, b and d would benefit from including quantifications and statistical analyses.”

Thank you for pointing out this issue. To address this point, we performed two more independent experiments and statistically compared the acetylated IMP2/total IMP2 ratio (Fig. 8a, b, and d and Supplementary Fig. 6b). As a result, we have now provided more convincing evidence. Again, we thank the reviewer for this important suggestion.

Report of Reviewer #2

Yoshizawa et al.

“SIRT7 serves as an energy-saving factor by suppressing brown fat thermogenesis”

This study by Yoshizawa et al. reports that SIRT7 has a role in the regulation of energy metabolism and thermogenesis through the regulation of UCP-1 protein levels in brown adipose tissue (BAT) in mice. They show that whole-body Sirt7 KO mice, as well as adipose tissue-specific and BAT-specific Sirt1 KO mice, maintain higher body temperature and energy expenditure compared to control mice under an ambient temperature. Whole-body Sirt7 KO mice also maintain their energy expenditure and body temperature during daily torpor and aging. The authors further dissected the molecular mechanism by which SIRT7 regulates UCP-1 protein levels and found that SIRT7 deacetylates insulin-like growth factor 2 mRNA binding protein 2 (IGF2BP2/IMP2), an RNA-binding protein that inhibits the translation of Ucp1 mRNA, at K438.

Although the results presented in this study look convincing and important, the biggest problem in this study is whether the phenotypes of body temperature and energy expenditure could be explained by the same mechanism in BAT. As described in detail below, the observed changes in UCP-1 protein levels in SIRT7 KO BAT are an unlikely explanation for increased energy expenditure. To rigorously address this critical issue, the authors should test their animals at thermoneutrality (~30C for mice). The IMP2-UCP-1 axis is most likely a reasonable explanation for increased body temperature under an ambient condition, and the authors might want to limit their presentation to this particular phenotype. But if that is the case, the novelty of the study would be dampened. Therefore, the authors are strongly encouraged to look deeper into the systemic interaction between adipose tissue and other tissues, such as the liver and skeletal muscle, to fully explain the mechanism for increased energy expenditure.

Major comments:

1) It has been known that the contribution of adipose tissue to whole-body energy expenditure is relatively very small and also that UCP-1 knockout mice show no differences in energy expenditure under an ambient condition. It is likely that increased BAT function through increased UCP-1 levels could provide a reasonable explanation for increased body temperature. Nonetheless, increased body temperature would not

necessarily lead to increased energy expenditure. Given significant increases in energy expenditure throughout light and dark times in whole-body, Ad, and BAd SIRT7 KO mice, it is very likely that other major organs, such as the liver and skeletal muscle, are involved in these observed changes in energy expenditure.

It would be ideal if the authors could conduct further analyses to obtain a more reasonable and convincing explanation for the energy expenditure phenotype. However, this may not be easy to do with a limited amount of time and resource for revision. Therefore, a minimal requirement would be to test whole-body and BAd SIRT7 KO mice at thermoneutrality. This particular experiment could provide more convincing results for the importance of SIRT7-dependent UCP-1 protein regulation in BAT.

2) For the same reason described above, it remains unclear whether “the importance of SIRT7 in the suppression of energy expenditure and thermogenesis is much higher in aged mice.” In aged mice, is there any change in SIRT7 protein amounts or SIRT7 activity?

Minor comments:

1) A quantification and scale bar should be provided in Fig. 1k.

2) The authors mentioned that VO₂, energy expenditure, and body temperature clearly declined in WT mice during aging in page 7, line 162-165. However, there is no statistical analysis for that. Those analysis should be performed in Fig. 2a-c.

3) Although a previous paper shows that Imp2 KO mice increase UCP-1 protein levels in BAT, it is worth checking UCP-1 mRNA and protein level in Sirt7 and Imp2 KO primary brown adipocytes in Fig. 6.

4) In Fig. 11, are there any clues why there is no difference in body temperature some time points, particularly during the dark time? How about UCP1 protein levels at those times? Although the authors do not show the oscillation of body temperature of adipose tissue- or brown adipocytes-specific Sirt7 KO mice (Fig. 3), it could be meaningful to compare the body temperature oscillation in these three KO models. Interestingly, there are less or no differences in VO₂ around ZT12 between WT and KO in each KO model mice (Fig. 1c, 31, 3f), similar to the oscillation of body temperature shown in Fig. 11. S

irt7 might have a role at specific time points, but not throughout a day. If UCP1 protein levels in BAT still showed a difference between WT and Sirt7 KO mice around ZT12, using an infrared camera would be a good idea to examine where the heat comes from.

5) Tables or figures should be provided for the mass spectrometry experiment including values and other candidates in Fig. 6.

6) Page 12, line 275-277, the authors mentioned a Western blotting experiment, however there is no result in Fig. 6.

Responses to the comments of Reviewer #2

We wish to thank the reviewer for the comment, “the results presented in this study look convincing and important.”, and for his/her constructive suggestions.

In response to your request, “the authors are strongly encouraged to look deeper into the systemic interaction between adipose tissue and other tissues, such as the liver and skeletal muscle, to fully explain the mechanism for increased energy expenditure.”, we have analyzed the expression of batokine genes in all three mouse models and of energy metabolism-related genes in the liver and muscle of *Sirt7* KO mice and performed the experiments at thermoneutrality in *Sirt7* KO and *Sirt7* BAdKO mouse lines. We have addressed all of the points raised by the three reviewers through new experiments and/or new text. Consequently, we have reorganized the text and figures to include the additional findings, particularly those concerning the systemic interaction between BAT and other tissues.

Major comments:

“1) It has been known that the contribution of adipose tissue to whole-body energy expenditure is relatively very small and also that UCP-1 knockout mice show no differences in energy expenditure under an ambient condition. It is likely that increased BAT function through increased UCP-1 levels could provide a reasonable explanation for increased body temperature. Nonetheless, increased body temperature would not necessarily lead to increased energy expenditure. Given significant increases in energy expenditure throughout light and dark times in whole-body, Ad, and BAd SIRT7 KO mice, it is very likely that other major organs, such as the liver and skeletal muscle, are involved in these observed changes in energy expenditure.

It would be ideal if the authors could conduct further analyses to obtain a more reasonable and convincing explanation for the energy expenditure phenotype. However, this may not be easy to do with a limited amount of time and resource for revision. Therefore, a minimal requirement would be to test whole-body and BAd SIRT7 KO mice at thermoneutrality. This particular experiment could provide more convincing results for the importance of SIRT7-dependent UCP-1 protein regulation in BAT.”

We greatly appreciate these recommendations. As suggested, we first analyzed the expression of energy metabolism-related genes and the protein levels of OXPHOS complexes I–IV in the liver and skeletal muscle of *Sirt7* KO mice. Although we could not detect obvious metabolic changes in the mRNA level (Supplementary Fig. 3d, e), it may

still be possible that SIRT7 in these organs regulates factors related to energy metabolism at the protein level. The protein levels of OXPHOS complexes I–IV were equivalent in the liver of WT and *Sirt7* KO mice, but complex I (NDUFB8) and IV (MTCO1) were decreased in the skeletal muscle of *Sirt7* KO mice (Supplementary Fig. 3f, g). Skeletal muscle might not contribute to the enhanced whole-body energy expenditure in *Sirt7* KO mice. We have added this information to page 10, paragraph 1.

Next, we performed the experiments at thermoneutrality in *Sirt7* KO and *Sirt7* BAdKO mouse lines. As shown in Fig. 3f and g, the phenotype of *Sirt7* KO mice comprising higher VO_2 and energy expenditure was remarkably attenuated at thermoneutrality but was still significantly higher during the active (dark) phase only. The body temperature of *Sirt7* KO mice was still slightly but significantly higher than that of WT mice (Fig. 3h). Therefore, we now conclude that SIRT7 suppresses energy expenditure and thermogenesis via multiple pathways, largely in BAT, but also in other organs. We have added this information to page 10, paragraph 1. In the case of *Sirt7* BAdKO mice at thermoneutrality, the phenotype of higher VO_2 , energy expenditure, and body temperature was almost completely abolished (Fig. 5f, g and Supplementary Fig. 5j). These results strengthen the conclusion that brown adipocytic SIRT7 suppresses whole-body energy expenditure and thermogenesis. We have added this information to page 12, paragraph 1.

Finally, we analyzed the gene expression of batokines and found that *Fgf21* and *Nrg4* in iBAT were highly elevated in KO types of all three lines (Fig. 3e, 4d, and 5d). These data suggest that brown adipocytic SIRT7 regulates whole-body energy expenditure through not only BAT thermogenesis, but also endocrine action. Unfortunately, it was hard to evaluate the contribution of the elevated *Fgf21* and *Nrg4* expression to whole-body energy expenditure because these elevated expressions were not seen at thermoneutrality (Supplementary Fig. 5k). We have added this consideration to the Discussion (from page 17, paragraph 2 to page 18, paragraph 1).

Consequently, we have substantially rewritten the manuscript to state that SIRT7 suppresses whole-body energy expenditure through multiple pathways, including BAT thermogenesis and endocrine action. We believe that the manuscript has been greatly improved by these modifications and we sincerely hope that you approve of our work.

“2) For the same reason described above, it remains unclear whether “the importance of SIRT7 in the suppression of energy expenditure and thermogenesis is much higher in aged mice.” In aged mice, is there any change in SIRT7 protein

amounts or SIRT7 activity?”

We apologize for our confusing explanation in the original manuscript. In contrast to genes activating BAT thermogenesis (*Prdm16*, *Ppargc1a*, *Ucp1*, and *Clstn3b*) that exhibited attenuated expression in iBAT of aged mice, the expression of *Sirt7* mRNA was steady, that is, the suppressor/activator ratio was higher in aged mice than in young mice. Thus, we believed that SIRT7 plays a more important role in aged mice. To avoid confusion, we have changed that sentence to “These results suggest that SIRT7 contributes more substantially to the suppression of energy expenditure and thermogenesis in aged mice.” (page 7, paragraph 1). Furthermore, we analyzed the protein level of SIRT7 in young and aged mice in response to your suggestion. Interestingly, SIRT7 protein was increased by aging (Supplementary Fig. 2f). We have added this information to page 7, paragraph 1.

Minor comments:

“1) A quantification and scale bar should be provided in Fig. 1k. ”

Thank you for pointing this out. We have added a quantification of the lipid area and a scale bar to Fig. 1k.

“2) The authors mentioned that VO₂, energy expenditure, and body temperature clearly declined in WT mice during aging in page 7, line 162-165. However, there is no statistical analysis for that. Those analysis should be performed in Fig. 2a-c.”

We apologize for the lack of statistical analysis of those data. We have performed the required statistical analysis for Fig. 2a–c.

“3) Although a previous paper shows that *Imp2* KO mice increase UCP-1 protein levels in BAT, it is worth checking UCP-1 mRNA and protein level in *Sirt7* and *Imp2* KO primary brown adipocytes in Fig. 6.”

We thank the reviewer for this comment. To address this request, we have analyzed the mRNA and protein levels of *Ucp1* in *Sirt7* and *Imp2* KO primary brown adipocytes. The UCP1 protein level was increased by IMP2 deficiency in WT cells, but not further increased in *Sirt7* KO cells (Fig. 7f), whereas *Ucp1* mRNA levels were not altered in any of the samples (Fig. 7g) (page 13, paragraph 2).

“4) In Fig. 1l, are there any clues why there is no difference in body temperature some time points, particularly during the dark time? How about UCP1 protein levels at those times? Although the authors do not show the oscillation of body temperature of adipose tissue- or brown adipocytes-specific *Sirt7* KO mice (Fig. 3), it could be meaningful to compare the body temperature oscillation in these three KO models. Interestingly, there are less or no differences in VO_2 around ZT12 between WT and KO in each KO model mice (Fig. 1c, 3l, 3f), similar to the oscillation of body temperature shown in Fig. 1l. *Sirt7* might have a role at specific time points, but not throughout a day. If UCP1 protein levels in BAT still showed a difference between WT and *Sirt7* KO mice around ZT12, using an infrared camera would be a good idea to examine where the heat comes from.”

Thank you for your comment. Although there is no evidence or clues, our hypothesis is the following. At the start of the active phase (ZT12), mice start to move and eat food and their body temperature reaches its maximal level due to physical activity- and diet-induced thermogenesis. In this saturated state, the relative contribution of BAT-UCP1 to whole-body thermogenesis is considered to be lower than that in the resting phase. Thus, the effect of an increased UCP1 protein level with *Sirt7* deficiency may be transiently masked by thermogenic pathways in other organs and UCP1-independent thermogenic pathways in BAT.

As suggested, we analyzed the body temperature oscillation in *Sirt7* AdKO and *Sirt7* BAdKO mice (Fig. 4c and 5c). In these two KO models as well, there were fewer or no differences in the body temperature at ZT14.

To address your question, we analyzed UCP1 protein levels in the iBAT of WT and *Sirt7* KO mice at around ZT12 (ZT14). The UCP1 protein level was still higher in *Sirt7* KO mice than in WT mice (Figure 1 for reviewer #2), suggesting that SIRT7 acts functionally even at ZT14. Because we do not have an infrared camera, we could not unfortunately examine where the heat comes from at around ZT12. We agree that study of the circadian role of SIRT7 is interesting. However, we have already provided considerable data and a study of the circadian rhythm would be a major undertaking that lies outside the scope of this study.

Figure 1 for reviewer #2.

“5) Tables or figures should be provided for the mass spectrometry experiment including values and other candidates in Fig. 6.”

Thank you for pointing this out. We have provided the mass spectrometry results, including values and other candidates (Supplementary Table 1).

“6) Page 12, line 275-277, the authors mentioned a Western blotting experiment, however there is no result in Fig. 6.”

As mentioned in our answer to Minor comment 3), we have checked the UCP1 protein level in *Sirt7* and *Imp2* KO primary brown adipocytes (Fig. 7f) (page 13, paragraph 2).

Report of Reviewer #3

The manuscript by Yoshizawa et al. describes a new mechanism of Sirt7 by deacetylating IMP2 at K438 to inhibit Ucp1 translation. This study started with thorough characterization of Sirt7-KO mice (2 independent lines in young and old ages) in body weight, energy expenditure, body temperature, food intake... etc. The authors then used adipocyte-specific (AdKO) and BAT-specific (BAdKO) Sirt7-KO to validate the essential role of Sirt7 in suppressing Ucp1 protein level and BAT thermogenesis. They further identified that the RNA-binding protein IMP2 as a substrate of Sirt7, whose deacetylation at K438 residue is important to inhibit Ucp1 translation. Overall, the main conclusions in this manuscript are well supported by the presented results from a combination of in vitro (cell culture and reporter assay) and in vivo (various global and conditional KO of Sirt7) experiments. The most part of this work is very solid and no major conceptual or experimental flaws, but there are some aspects with room for improvement as described below.

Specific Comments:

- 1) In the introduction- line 90, about posttranscriptional regulation of Ucp1 mRNA, 2 more papers, Takahashi et al 2015, Cell Rep. 13:2756 and Chen et al 2018 EMBO J 37:e99071, should be included.
- 2) Lines 102-105, “SIRT7 is a relatively unique sirtuin. Its enzymatic activities and functional roles had been only partially elucidated until recently, but recent studies have revealed many biological functions for SIRT7,” What does it mean about the relative uniqueness of Sirt7 in comparison to other sirtuins? “had been only partially elucidated until recently, but recent studies.....” is an awkward expression, please rephrase it.
- 3) In lines 117-118, “We also demonstrate acetylation-dependent of Ucp1 mRNA by RNA-binding proteins (RBP).” Because only IMP2 was studied here, “RNA-binding proteins” refer to? Any other RNA-binding proteins, such as CPEB2, BRF1 or those listed in the Supp Fig 5d, were identified from Halo-tag-Sirt7 pull-down experiments (Fig. 6a)?
- 4) The authors appeared to generate floxed Sirt7 allele in C57BL/6J mice but they crossed the floxed mice with adipoq-cre (JAX010803) and Ucp1-cre (JAX024670) mice in FVB/N background. Because genetic background can influence various metabolic parameters, how does the authors control the variations caused by mixed genetic

background?

5) In Fig 4b, 4f, 4g, the immunoblotting of Sirt7 should be included. It is important to show the depletion of Sirt7 in iBAT.

6) The previous study (EMBO J 37:e99071) reported that alternative polyadenylation produces Ucp1 transcripts carrying long (~10%) or short (~90%) 3'-UTR in mouse BAT. Does deacetylation/acetylation of IMP2 affect translation of both long and short forms of Ucp1 transcripts? In Fig 7e, which kind of 3'-UTR was used for the reporter assay, the majority short form or the minority long form? The authors should test IMP2 and its deacetylation/acetylation-mimetic mutants in the luciferase reporters appended with long and short Ucp1 3'-UTR. Moreover, they should also demonstrate that K438R but not K438Q mutant binds to Ucp1 mRNA. Please also specify which Ucp1 3'-UTR (long, short or both) was bound by IMP2.

Responses to the comments of Reviewer #3

We wish to thank the reviewer for the comment “The most part of this work is very solid and no major conceptual or experimental flaws.”, and for his/her constructive suggestions.

In response to your comments, we have performed an RNA immunoprecipitation assay using long and short *Ucp1* 3'-UTR. We have addressed all the points raised by the three reviewers through new experiments and/or new text. Consequently, we have reorganized the text and figures to include the additional findings, particularly those concerning the systemic interaction between BAT and other tissues.

Specific Comments:

“1) In the introduction- line 90, about posttranscriptional regulation of Ucp1 mRNA, 2 more papers, Takahashi et al 2015, Cell Rep. 13:2756 and Chen et al 2018 EMBO J 37:e99071, should be included.”

We apologize for overlooking these citations. We have now included these studies in the Introduction (page 4, paragraph 2) (ref. 23 and 24).

“2) Lines 102-105, “SIRT7 is a relatively unique sirtuin. Its enzymatic activities and functional roles had been only partially elucidated until recently, but recent studies have revealed many biological functions for SIRT7,” What does it mean about the relative uniqueness of Sirt7 in comparison to other sirtuins? “had been only partially elucidated until recently, but recent studies.....” is an awkward expression, please rephrase it.”

We apologize for our incomprehensible explanation in the original manuscript. The reasons why we consider SIRT7 to be a relatively unique sirtuin are as follows: 1) *Sirt7* KO mice are resistant to HFD-induced obesity, glucose intolerance, and fatty liver, in striking contrast to other reported *sirtuin* KO mice³⁵; 2) *Sirt7* KO mice exhibit increased body temperature and whole-body energy expenditure under HFD conditions, in contrast to other *sirtuin* KO mice³⁵; 3) SIRT1 and SIRT7 control adipogenesis in opposite directions (Picard et al., *Nature* 2004); 4) only SIRT7 is present in the nucleolus as well^{33,34}; 5) SIRT7 targets very limited histone modifications compared with other nuclear sirtuins^{33,34}; 6) DNA and RNA activate specifically SIRT7 (Tong et al., *ACS Chem. Biol.* 2016 and 2017).

To avoid any misunderstanding, we have removed the sentence concerning the

uniqueness and rephrased the awkward expression as follows: “Until recently, SIRT7 had been the least studied sirtuin, but it has lately been reported to have many important biological functions, with roles in ribosome biogenesis, the stress response, genome integrity, metabolism, cancer, and aging^{33,34}.” (page 5, paragraph 2).

References

Picard, F. et al. Sirt1 promotes fat mobilization in white adipocytes by repressing PPAR-gamma. *Nature* **429**, 771-776 (2004).

Tong, Z. et al. SIRT7 is activated by DNA and deacetylates histone H3 in the chromatin context. *ACS Chem. Biol.* **11**(3), 742-747 (2016).

Tong, Z. et al. SIRT7 is an RNA-activated protein lysine deacylase. *ACS Chem. Biol.* **12**(1), 300-310 (2017).

“3) In lines 117-118, “We also demonstrate acetylation-dependent of Ucp1 mRNA by RNA-binding proteins (RBP).” Because only IMP2 was studied here, “RNA-binding proteins” refer to? Any other RNA-binding proteins, such as CPEB2, BRF1 or those listed in the Supp Fig 5d, were identified from Halo-tag-Sirt7 pull-down experiments (Fig. 6a)?”

We apologize for our inaccurate explanation in the original manuscript. We have limited this sentence to the IMP2 as follows: “We also demonstrate acetylation-dependent regulatory mechanisms attenuating the translation of *Ucp1* mRNA by the RNA-binding protein (RBP) IMP2.” (page 5, paragraph 3).

“4) The authors appeared to generate floxed Sirt7 allele in C57BL/6J mice but they crossed the floxed mice with adipoq-cre (JAX010803) and Ucp1-cre (JAX024670) mice in FVB/N background. Because genetic background can influence various metabolic parameters, how does the authors control the variations caused by mixed genetic background?”

Thank you for pointing this out, and we apologize for omitting the important information. We obtained these mice from Dr. Evan Rosen, who donated it to the Jackson Laboratory. They had already back-crossed these mice for over eight generations with C57/BL6J mice^{48,49}. Thus, these mice are almost in a C57/BL6J background. We have added this information to page 19, paragraph 2.

“5) In Fig 4b, 4f, 4g, the immunoblotting of Sirt7 should be included. It is important to show the depletion of Sirt7 in iBAT.”

We thank the reviewer for this comment. To address this request, we have analyzed SIRT7 protein levels in *Sirt7* KO, *Sirt7* AdKO, and *Sirt7* BAdKO mice to show the depletion of SIRT7 in iBAT. As shown in Fig. 4f and 5e, the depletion of SIRT7 was not perfect in *Sirt7* AdKO mice and was more inefficient in *Sirt7* BAdKO mice. Therefore, we have added the following consideration to the Discussion (page 17, paragraph 1): “Moreover, it is probable that a variation in the depletion efficiency of SIRT7 by different Cre drivers (Fig. 4f, 5e) affected these observed changes in gene expression.”

“6) The previous study (EMBO J 37:e99071) reported that alternative polyadenylation produces Ucp1 transcripts carrying long (~10%) or short (~90%) 3'-UTR in mouse BAT. Does deacetylation/acetylation of IMP2 affect translation of both long and short forms of Ucp1 transcripts? In Fig 7e, which kind of 3'-UTR was used for the reporter assay, the majority short form or the minority long form? The authors should test IMP2 and its deacetylation/acetylation-mimetic mutants in the luciferase reporters appended with long and short Ucp1 3'-UTR. Moreover, they should also demonstrate that K438R but not K438Q mutant binds to Ucp1 mRNA. Please also specify which Ucp1 3'-UTR (long, short or both) was bound by IMP2.”

We greatly appreciate these suggestions. To analyze the binding between IMP2 and *Ucp1* 3'-UTR, we performed an RNA immunoprecipitation assay (Method: page 29, paragraph 1 and 2). First, we verified that IMP2 naturally binds to the long form, but not to the short form (Supplementary Fig. 6d). Therefore, we analyzed the binding of IMP2^{WT}, IMP2^{K438R}, and IMP2^{K438Q} to long *Ucp1* 3'-UTR. The binding of IMP2^{K438Q} to long *Ucp1* 3'-UTR was significantly low compared with IMP2^{WT} and IMP2^{K438R} (Fig. 8e). We have added this information to the text from page 14, paragraph 3 to page 15, paragraph 1. In the original Fig 7e (new Fig. 8f), we used only long *Ucp1* 3'-UTR for the reporter assay, because IMP2 does not bind to the short form.

REVIEWERS' COMMENTS

Reviewer #1 (Remarks to the Author):

The authors have addressed our concerns and we now consider the manuscript suitable for publication.

Minor comments:

- In Figure 3a, is *Cebpb* expression significantly altered? Please, check if the mistake is in the text or in the graph.
- There is a typo in Figure 3b and others: *Clstn3b* and not *Clsta3b*.
- There is a typo in Figure 5a: Are the animals in this panel BAdKO or AdKO?

Reviewer #2 (Remarks to the Author):

In this revised manuscript, the authors adequately addressed all criticisms and questions from this reviewer. Their conclusion is now significantly strengthened, and therefore, the manuscript is ready for the acceptance in Nature Communications. Congratulations!

Reviewer #3 (Remarks to the Author):

The authors have properly addressed my concerns and I thank them for their effort to perform additional experiments to clarify my questions.

Report of Reviewer #1

The authors have addressed our concerns and we now consider the manuscript suitable for publication.

Minor comments:

- In Figure 3a, is *Cebpb* expression significantly altered? Please, check if the mistake is in the text or in the graph.
- There is a typo in Figure 3b and others: *Clstn3b* and not *Clsta3b*.
- There is a typo in Figure 5a: Are the animals in this panel BAdKO or AdKO?

Responses to the comments of Reviewer #1

We wish to thank the reviewer for the acceptance. Your suggestions substantially changed our study and improved our manuscript. Again, we greatly appreciate the reviewer for fair peer review.

Minor comments:

"In Figure 3a, is *Cebpb* expression significantly altered? Please, check if the mistake is in the text or in the graph."

We apologize for our mistake in the original text. *Cebpb* expression was not altered in *Sirt7* KO mice (Fig.3a), but significantly changed in *Sirt7* BAdKO mice (Fig. 5d). We have corrected this error in the text (*Cebpb* was removed: page 8, paragraph 2 and *Cebpb* was added: page 11, paragraph 2).

"There is a typo in Figure 3b and others: *Clstn3b* and not *Clsta3b*."

We apologize for our typo in Figures. We have corrected this typo in Fig.3b, 4d, 5d, 6b.

"There is a typo in Figure 5a: Are the animals in this panel BAdKO or AdKO?"

We apologize for our typo in Fig.5a. The animals in Fig.5a are *Sirt7* BAdKO mice. We have corrected this mistake.

Report of Reviewer #2

In this revised manuscript, the authors adequately addressed all criticisms and questions from this reviewer. Their conclusion is now significantly strengthened, and therefore, the manuscript is ready for the acceptance in Nature Communications. Congratulations!

Responses to the comments of Reviewer #2

We wish to thank the reviewer for the acceptance. Important new results according to your judicious requests greatly improved our manuscript. Again, we appreciate the reviewer for fair peer review.

Report of Reviewer #3

The authors have properly addressed my concerns and I thank them for their effort to perform additional experiments to clarify my questions.

Responses to the comments of Reviewer #3

We appreciate the reviewer for the fair peer review and acceptance. Your suggestions greatly improved our manuscript.